

# Mātauranga Māori in geomorphology: existing frameworks, case studies and recommendations for Earth scientists

Clare Wilkinson[1], Daniel C.H. Hikuroa[2], Angus H. Macfarlane[3], and Matthew W. Hughes[4]

[1]School of Earth and Environment, University of Canterbury, Christchurch, 8140, New Zealand
[2]Department of Māori Studies, University of Auckland, Auckland, 1142, New Zealand
[3]College of Education, Health & Human Development, University of Canterbury, Christchurch, 8140, New Zealand
[4]Civil & Natural Resources Engineering, University of Canterbury, Christchurch, 8140, New Zealand

*Correspondence to:* Clare Wilkinson (clare.wilkinson@pg.canterbury.ac.nz)

**Abstract.** Mixed-method bicultural research in Aotearoa New Zealand, including the weaving of Indigenous and other
knowledges, is experiencing a resurgence within many academic disciplines. However, mātauranga Māori–the knowledge, culture, value and worldview of the Indigenous peoples of Aotearoa New Zealand—and Te Ao Māori, the Māori world, is poorly represented within geomorphological investigations. Here, we review existing efforts to include Indigenous knowledge in geologic and geomorphic studies from the international research community and provide an overview of the current state of mātauranga Māori within research endeavours in Aotearoa New Zealand. We review three theoretical frameworks for
including mātauranga Māori in research projects and three models for including Māori values within research. We identify direct benefits to geomorphology and discuss how these frameworks and models can be adapted for use with Indigenous knowledge systems outside of Aotearoa New Zealand. The aim of this review is to encourage geomorphologists around the world to engage with local Indigenous peoples to develop new approaches to geomorphic research. In Aotearoa New Zealand, we hope to inspire geomorphologists to embark on research journeys that engender genuine partnership with Māori and that
promote toitū te mātauranga, the enduring protection, promotion and respect of mātauranga Māori.

Keywords: geomorphology, mātauranga Māori, bicultural research

## 1 Introduction

Earth scientists are increasingly recognising the benefits of conducting mixed-methods bicultural research (e.g., Townsend et
al., 2004; Tipa, 2009; Harmsworth et al., 2011; Crow et al., 2018; Hikuroa et al., 2018). Oral histories, lore and mythologies from Indigenous communities, explained through their respective worldviews, frequently feature stories of geomorphic or landscape change in their tribal lands (e.g., Gottesfeld et al., 1991; McMillan and Hutchinson, 2002; Hikuroa, 2017). Indigenous knowledge and, in particular, oral histories, have been shown to complement scientific endeavours by detailing specific natural events that were otherwise poorly understood or documented by scientists (e.g., Swanson, 2008; King and
Goff, 2010; Reid et al., 2014; Nunn and Reid, 2016) and fill knowledge gaps that science cannot (Bohensky and Maru, 2011).





As such, Indigenous knowledge can provide an observational starting point, or corroborative evidence, for scientific investigations.

Historically, there has been discord between the scientific and Indigenous knowledge epistemologies. The science community has traditionally considered Indigenous knowledge systems and oral histories unreliable, inaccurate, untruthful, and doubtable (Durie, 2004). Until quite recently, anthropologists still promoted the unreliability of unwritten (i.e., oral) legends that refer to events more than 1000 years before present (Simic, 2002, as cited in Reid et al., 2014). On the other hand, Indigenous communities have frequently expressed opposition to science due to its inertia to recognise nature as something more than a controllable, testable, and exploitable medium (Smith, 1999; Hikuroa et al., 2011). While scientists are typically detached "observers" and analysers of natural systems (Cruikshank, 2012; Hikuroa, 2017), Indigenous communities position themselves within an extended genealogy that considers nature as kin (Suzuki and Knudtson, 1992; Salmón, 2000). In Indigenous worldviews and knowledge systems, humans are active participants within natural systems (Hikuroa, 2017; Pingram et al., 2019). Tensions between Indigenous knowledge and science—particularly tensions around rigor of knowledge generation, credibility, worldview, and ability to be evaluated—have created challenges for integrating knowledge systems in the past (Mercier, 2007; Bohensky and Maru, 2011).

Until recently, the historic discord between science and Indigenous knowledge prevented the synergies that do exist between the two knowledge systems from advancing new understandings. In the past 10-15 years, a resurgence of sincere, respectful and reciprocal re-engagement between scientific and Indigenous communities has generated multiple national and international guiding policies for genuinely transformative approaches to research (e.g., Hīkina Whakatutuki Ministry of Business, Innovation and Employment, n.d.; Ministry of Research, Science and Technology, 2007; UN General Assembly, 2007). Re-engagement has identified research needs and aspirations of both Indigenous communities and scientists, leading to co-creation and co-development of research projects with respective responsibilities clearly defined. In 2007, the United Nations Declaration on the Rights of Indigenous Peoples (UNDRIP) catalysed reconsideration and rebalancing of Indigenous peoples' rights (Hikuroa et al., 2018). The UNDRIP formalised obligations of participating governments to support and protect Indigenous communities' rights to maintain cultural heritage, traditional knowledge, expression of their sciences, oral traditions and technologies (UN General Assembly, 2007), and created a platform on which mixed-methods research can be formulated, discussed and carried out. To date, legal and constitutional initiatives that build upon UNDRIP policies and establish the "rights of nature" have occurred in Bolivia, India, New Zealand, Australia and Ecuador (Boyd, 2017; Brierley et al., 2018; O'Donnell and Talbot-Jones, 2018). Though these advances and recognitions are most prevalent in the policy sphere, they are transferrable to scientific research and have, in a few cases, acted as guidelines for culturally responsible and respectful research at the interface of Indigenous knowledge and Western science.

The international geosciences community is increasingly demonstrating interest in Indigenous knowledge systems and participation with Indigenous groups (e.g., Tipa, 2009; King and Goff, 2010; Harmsworth et al., 2011; Harmsworth and Roskruge, 2014; Pardo et al., 2015; Riu-Bosoms et al., 2015; Nunn and Reid, 2016; Hikuroa, 2017; Brierley et al., 2018; Crow et al., 2018). Indigenous knowledge has been used to define research needs in geospatial research projects (e.g., Poole and



Biodiversity Support Program, 1995 *as cited in* Pacey, 2005; Harmsworth, 1999; Alessa et al., 2011; Te Rūnanga o Ngāi Tahu, 2019), natural hazard research (Swanson, 2008; Goff et al., 2010; King and Goff, 2010; King et al., 2018), natural hazard risk reduction planning (Cronin et al., 2004; Becker et al., 2008; Walshe and Nunn, 2012; Rumbach and Foley, 2014; Hiwasaki et al., 2014; Pardo et al., 2015; Rahman et al., 2017), climate-change resilience (Cruikshank, 2001, 2012; Ford and Smit, 2004; Janif et al., 2016; Iloka, 2016), environmental management (Londono et al., 2016), soil classification (Oudwater and Martin,

2003; Harmsworth and Roskruge, 2014) and geomorphology/hydrology research (Londono et al., 2016; Hikuroa, 2017). Moreover, Indigenous place names commonly indicate knowledge of landscape features and geomorphology (Carter, 2005; Kharusi and Salman, 2015; Riu-Bosoms et al., 2015; Atik and Swaffield, 2017). Thus, culturally responsible and respectful weaving of Indigenous knowledge into Earth science has the potential to corroborate, bolster and create knowledge.

This review focuses on recent efforts to include mātauranga Māori (Māori Indigenous knowledge) alongside
geomorphology in research conducted within Aotearoa New Zealand (henceforth Aotearoa-NZ). Although Aotearoa is a Māori name for New Zealand's North Island, to reflect the nation's bi-cultural foundation it is commonly used in this context (e.g. Aotearoa-NZ) to mean all of New Zealand. Mātauranga Māori can be described as a detailed and complex knowledge system originating from Māori ancestry (Paul-Burke et al., 2018), including culture, values and Māori worldview (Hikuroa, 2017). This review begins with a discussion of international efforts in mixed-methods research at the interface of Indigenous

knowledge and geoscience, concluding with a focus on geomorphology. We then introduce Te Ao Māori (the Māori world), discuss obligations of the New Zealand government to Māori, and present frameworks for conducting mixed-methods scientific research with iwi and hapū (tribes and family groupings—the principle political units with whom scientists engage) in Aotearoa-NZ in this space. We then provide case studies of framework development and recommendations for framework implementation in geomorphology research. Finally, we provide direct examples of including Indigenous knowledge in

geomorphic research and discuss how the frameworks and models reviewed here can be applied outside of the Aotearoa-NZ context. We believe that the scientific world may learn some valuable lessons from Aotearoa-NZ about how Indigenous knowledge and geomorphology can work together to create new and innovative understandings about how to live with and learn about Earth surface systems.

The authors assert that there is no expectation that mātauranga be given away by iwi and hapū to scientists and
acknowledge that the mātauranga presented here is not our own, and that we have gained approval through the Human Ethics Committee at the University of Canterbury (Christchurch, NZ) to conduct this research. Scientists cannot rebuild or revitalise mātauranga; that is for Māori to do (Broughton et al., 2015). Māori have been leading revitalisation projects for over 30 years (Broughton et al., 2015), and Māori values and knowledge are being increasingly included in ecology and resilience studies. We uphold that the geoscience community is primed to contribute to further reinvigoration of mātauranga by welcoming it

alongside science for greater understanding of Earth surface phenomena. Our intentions for this review are to encourage inclusion of Indigenous knowledge and values for guiding scientific research. We herein acknowledge the mana whenua of Aotearoa-NZ as the rightful holders of mātauranga.



## 2 Overview of international research at the interface of Indigenous knowledge and science

Evaluating events recorded in Indigenous peoples' oral histories with scientifically-investigated landforms or processes is not
a new concept. Gottesfeld et al. (1991) examined a Holocene debris flow near Hazelton, British Columbia (ca. 3500 BP, before
present) and discussed how the event could be the same as a story belonging to the local Indigenous peoples, the Gitksan, of
the Medeek, a devastation-wreaking grizzly bear that charged down the mountain, uprooting trees and leaving a wide gash in
the hillside. Scientists have dated the debris flow to a time when the Gitksan people occupied the area. Given that both accounts
describe the same event, and with scientific dating aligning with oral history of Gitksan presence in the area, it is likely that
both scientists and the oral history can contribute observations and knowledge about the event. Similarly, Eisbacher and Clague
(1984) discussed Indigenous perspectives of debris flows in the European Alps, wherein the events were described as "…raging
giants and infuriated dragons" that were responsible for "sudden roar[s] in the gorges and the violent eruption of rubbly debris
onto fields and communities" (p. 74). More recently, scientists have recognised the plethora of land- and seascape terms within
Indigenous languages (e.g., O'Connor and Kroefges, 2008; Senft, 2008) and the wealth of information about dynamic Earth
processes stored in Indigenous place names (Kharusi and Salman, 2015; Riu-Bosoms et al., 2015; Atik and Swaffield, 2017).
For example, Senft (2008) indicated that the peoples of Kaile'una Island (Papua New Guinea) have specific terms for the sea
at different points along a reef barrier. *O tulupwaka* means the 'sea between the inner and outer reef'; *omata sulusulu* means
'sea that covers the outer reef'; *omata takivi* means 'sea between the drop-off of the outer reef and the deep sea'; and *o
tulubwabwau* means the 'deep dark sea' (Senft, 2008). Similarly, Barrera-Bassols (2015) described a geomorphic map created
by the Purhepecha peoples of central Mexico based on Indigenous soil classification names that shows similarities to
scientifically-generated relief maps, though different criteria were used to create the maps. Barrera-Bassols (2015) showed that
the Purhepecha peoples have a geomorphic soil classification system that correlates strongly with scientific approaches to soil
classification, where maps of soil distribution generated by locals using local knowledge are similar to soil maps created by
scientists. Others (e.g. Payton et al., 2003; Hillyer et al., 2006) have noted similar results in other parts of the world.

The international science community has also learnt from oral histories for geologic hazard research. Swanson (2008)
showed that native Hawai'ian oral traditions involving the volcano goddess Pele record a detailed understanding of the Kīlauea
volcanic system's eruptive history over the past 400 years. The timeline of volcanic eruptions held in oral histories aligns with
scientific analysis of the volcano's eruptive history. Thus, the oral traditions accurately recorded and described geologic events
(Swanson, 2008). Because of the growing recognition of oral traditions as place-based repositories of accurate geologic
information, the scientific community is increasingly working with Indigenous groups to elucidate natural hazards. As a result,
volcanic hazard management schemes that include elements of local Indigenous knowledge and Western science-based
management have been developed in Vanuatu (Cronin et al., 2004) and Papua New Guinea (Mercer and Kelman, 2010).
Indigenous knowledge and perspectives have also been used in tsunami hazard management plans in Vanuatu (Walshe and
Nunn, 2012), the Pacific Northwest of the United States of America (Becker et al., 2008), Indonesia (Hiwasaki et al., 2014;
Rahman et al., 2017), the Chatham Islands (Thomas, 2018), the Philippines (Hiwasaki et al., 2014), and Samoa (Rumbach and



Foley, 2014). There is even more research discussing integration of Indigenous knowledge and Western science for disaster risk reduction (e.g., Mercer et al., 2007, 2010; Kelman et al., 2012), but this is outside the scope of this review.

Indigenous knowledge is also being used to better understand climate-change, seasonal climate forecasts and climate-change resilience guidelines. Janif et al. (2016) reported that in Fiji, stories held by Indigenous locals of catching certain types
of fish can indicate changes in sea surface temperatures. Similarly Cruikshank (2012) described stories of salmon migration (or lack thereof) held by Indigenous Alaskans that provided insight into glacial activity during the Little Ice Age (1550-1850 CE, common era). Their stories reflect that though climate change may be a global phenomenon, it has extremely local effects. Nyong et al. (2007) also demonstrated that local solutions to global climate change effects can bring great benefits to climate-change resilience plans. In West Africa Sahel, the ancestors of many Indigenous populations have experienced and adapted to
historic climate extremes that surpassed those predicted by current International Panel for Climate Change (IPCC) models (Nyong et al., 2007). Iloka (2016) also recognised that Indigenous communities in Africa have a wealth of environmental knowledge, passed on by previous generations who endured and survived climate conditions far more extreme than current predictions. Therefore, mitigation strategies developed by previous generations may have implications for future solutions.

Research that explicitly includes geomorphic techniques alongside Indigenous knowledge is not as abundant in
academic literature as research that incorporates Indigenous knowledge and values into ecology (e.g., Rainforth and Harmsworth, 2019) or disaster risk reduction research. Many publications have shown the potential for conducting geomorphic research with native peoples, evidenced by the large amount of studies investigating Indigenous languages for landscape, geomorphic, pedologic, hydrologic and glacial terms or classification schemes (e.g., Payton et al., 2003; Hillyer et al., 2006; O'Connor and Kroefges, 2008; Senft, 2008; Kharusi and Salman, 2015; Riu-Bosoms et al., 2015, p.; Barrera-Bassols, 2015;
Atik and Swaffield, 2017). We recognise that geomorphic analysis with Indigenous communities could feature in studies covering ecology and biology because Indigenous peoples do not separate ecosystems from landscapes, but there is a dearth of purely geomorphic studies that aim to weave Indigenous knowledge with science. Bohensky and Maru (2011) provide an extensive review of Indigenous knowledge and Western science integration in the resource management field, but, again, is largely focused on ecology. Some studies that do explicitly address geomorphic research with Indigenous communities
typically cover hydrologic and environmental management (e.g., Londono et al., 2016) or soil classification (e.g., Barrera-Bassols, 2015). To our knowledge, most studies that explicitly incorporate Indigenous knowledge and values alongside geomorphic research have been conducted in Aotearoa-NZ, and are the focus of the remainder of this review.

## 3 Mixed-method geoscience research in contemporary Aotearoa-NZ

### 3.1 Te Ao Māori (the Māori worldview)

Te Ao Māori has, at its foundation, relationships between everything seen and unseen, humans and more-than humans, the natural and beyond-natural world, and in turn, shapes Māori ways of doing and living (Clapcott et al., 2018). Māori have been creating and revising their mātauranga since they first arrived to Aotearoa-NZ many centuries years ago (Hikuroa, 2017). After





Table 1: Glossary of Māori terms (as used in this paper)

| | |
|---|---|
| Arohatanga | Care, respect, love |
| Atua | Departmental gods |
| Hapū | Sub-tribe |
| Hine-Titama | The first human, a woman |
| Io-Matua-Kore | The supreme 'first' being in Māori cosmology |
| Iwi | Tribe |
| Kaitiaki/kaitiakitanga | Guardian and the act of guardianship |
| Ki Uta Ki Tai | Concept expressing the importance of catchments extending from the mountains to the sea |
| Mahinga kai | Traditional food gathering practices and places |
| Mana | Authority, prestige |
| Mana whenua | People with authority over the land |
| Manaakitanga | Acts of caring for and giving |
| Māramatanga | Enlightenment, understanding, a phase in which knowledge can be applied |
| Mātauranga Māori | Knowledge held by Māori, the Indigenous peoples of Aotearoa New Zealand |
| Mātauranga-a-iwi | Iwi-specific (tribal) knowledge |
| Mauri | Life force, essence |
| Mōhiotanga | Acknowledgement, respect, awareness of potential |
| Pākehā | Non-Māori (European descent) New Zealander |
| Papatuanuku | Earth mother |
| Pūrākau | Oral record or history |
| Ranginui | Sky father |
| Rūnanga | Tribal council or governing board |
| Tane | God of the forests; created the first human |
| Taniwha | Supernatural creatures in Māori legends, often taking the form of a serpent or water monster |
| Tangata whenua | People of the land |
| Taonga | Treasure (noun), to be treasured (verb) |
| Te Ao Māori | Māori worldview |
| Te Ao Marama | The world of light, the world we inhabit |
| Te Kore | The nothingness, the potential for life |
| Te Po | The darkness, the night |
| Te taiao | The natural world; the environment |
| Tikanga | Customary practices, values, protocols |
| Tino rangatiratanga | Self-determination |
| Ūkaipō | Roots |
| Wairuatanga | Spiritual dimension |
| Whakakotahitanga | Respect for differences, ability to reach consensus, participatory inclusion in decision-making |
| Whakapapa | Ancestral genealogy, applicable to all elements of nature |
| Whānau | Family or close kin network |
| Whānaungatanga | Family connections |

settling, Māori formed distinct groups (about 40 iwi and hundreds of hapū), all of which built their identity from the surrounding mountains, lakes and rivers (Ruru, 2018). These tribal identities have implications for mātauranga-

a-iwi (iwi-specific mātauranga), tribal ancestry, credibility and iwi-specific guardianship of tribal lands. Glossaries of Māori kupu, or words (Table 1), and key English terminologies used in this paper (Table 2) are provided for reference.





Table 2: Glossary of English terms (as used in this paper)

| | |
|---|---|
| Cultural association | The cultural uses and values associated with a landscape |
| Framework | Theoretical guides to research; methodologies |
| Geomorphic rights | Rights of a river with the status of legal personhood, understood from a geomorphic perspective |
| Indigenous knowledge | Knowledge generated by Indigenous peoples using Indigenous methods and usually including values, culture and worldview |
| Knowledge | Intellectual capital generated over time and carried through a range of channels including stories, songs, philosophies and teachings |
| Method | Acts by which research is conducted |
| Methodology | Philosophical approach to research |
| Model | Actionable guides to research; methods |
| Science | The pursuit of knowledge according to the scientific method |
| Treaty of Waitangi | Official founding document of Aotearoa-NZ that joined Māori chiefs with the British Crown in 1840 |
| Value | Guiding principles that support or enable acceptable actions |

**3.1.1 Whakapapa and tikanga—validity through ancestry**

Whakapapa is the Māori way of understanding the world through genealogies (Forster, 2019). It links people to flora, fauna, mountains, rivers, oceans and lakes through an understanding that all of nature is descended from the atua (Fig. 1; Harmsworth and Awatere, 2013; Forster, 2019). Whakapapa informs tikanga, or cultural protocols and habits, which in turn informs how one should conduct their life (Graham, 2009).



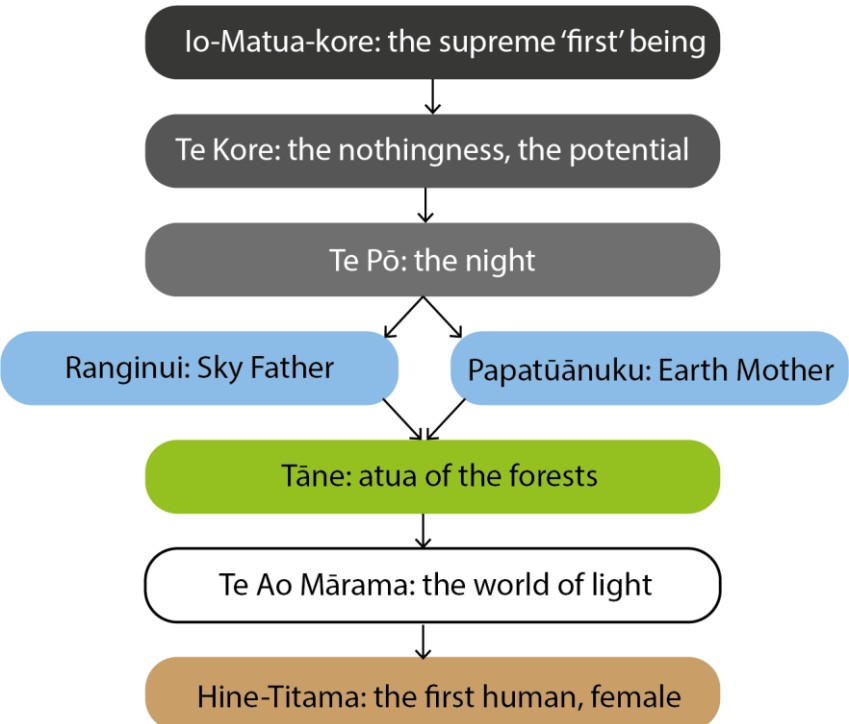

**Figure 1. The pedigree of mankind in Te Ao Māori.** *Modified from* **Graham, 2009.**

Whakapapa is at the core of Indigenous Māori knowledge generation (Graham, 2009). Whakapapa legitimates Māori epistemologies within research and fosters credibility by establishing connections between researchers and subjects, and by guiding research questions based on tikanga (Graham, 2009). By understanding that all things—both physical and metaphysical—are connected through genealogy (Hikuroa, 2017), it can be understood that whakapapa is a structured methodology for creating mātauranga (Royal, 1998). The relationships within whakapapa inform histories, stories, and

interactions, which can be analysed in a Māori-centred way to create new knowledge (Fig. 2).

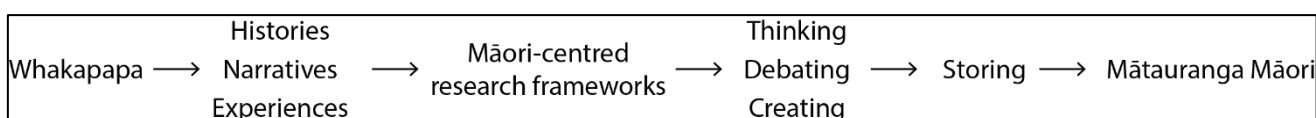

**Figure 2. Generation of Māori knowledge.** *Modified from* **Graham, 2009.**

### 3.1.2 Mātauranga Māori

Mātauranga Māori is a detailed and dynamic way of knowing that has its ūkaipō in Māori ancestry (Paul-Burke et al., 2018). Mātauranga is a taonga that is lived, practiced, tested, updated and that grows and develops as it is passed from generation to





generation. Based on Polynesian origins (Clapcott et al., 2018), Māori have been developing their mātauranga since their
arrival to Aotearoa-NZ some 800-1000 years ago (Broughton et al., 2015). Mātauranga is not only knowledge, but is also a
method of expressing knowledge through language, cultural practices, values, principles and ethics (Hikuroa, 2017; Paul-
Burke et al., 2018). Mātauranga taiao, or Māori environmental knowledge, is both traditional and contemporary, and reflects
the totality of Māori experiences interacting with the environment during their occupation of Aotearoa-NZ (King et al., 2007).

Mātauranga-a-iwi provides local, place-based knowledge for an iwi's tribal area. This knowledge can provide intimate
understandings of landscape dynamics and change through time. Mātauranga-a-iwi is informed directly by whakapapa, because
local landscape features are seen as kin through genealogical ties (Wilkinson and Macfarlane, in press; Ruru, 2018). The aim
is to live with the environment in an intergenerationally sustainable way, since landscapes are part of the ancestry, in which
the landscape and its resources are respected as elders. Interacting with specific landscapes has generated and developed
mātauranga-a-iwi, and continues to refine local Indigenous knowledge.

### 3.1.3 Kaitiakitanga

In Te Ao Māori, mana whenua are the kaitiaki of their lands and waters. They are the guardians of the physical and cultural
environments. Kaitiakitanga is a responsibility to maintain the well-being of people and environment. Contemporary
kaitiakitanga can be understood as implementation of mātauranga-informed decisions and management (Clapcott et al., 2018;
Paul-Burke et al., 2018). It can also be understood as the responsibility to guide research priorities in the interest of the
environment and the landscape. For example, studies that consider water quality and quantity and establish baseline minimums
for flow (e.g., Tipa, 2009b; Crow et al., 2018; Hikuroa et al., 2018) are expressions of kaitiakitanga in modern research and
management.

### 3.2 Obligations of the Aotearoa New Zealand government to Māori

### 3.2.1 The Treaty of Waitangi

The Treaty of Waitangi is the founding document of the modern state of Aotearoa-NZ (Hudson and Russell, 2009). The Treaty
represents the establishment of a formal relationship between the British Crown and Māori, in which Māori are legal partners
of the Crown. Two versions of the text exist: one in te reo Māori and one in English. The te reo Māori text was signed in
Waitangi on 6 February 1840 by more than 40 Māori chiefs, and was then circulated to other Māori communities around the
country (Anderson et al., 2015). Not all chiefs signed the Treaty, but it did receive more than 500 Māori signatures. The Treaty
established that Māori taonga—including mātauranga—would be protected and that Māori had the right to participate as active
citizens of Aotearoa-NZ. To Māori, the status of the Treaty remains as strong and relevant today as it did in 1840; however,
the applicability of the Treaty within modern Aotearoa-NZ has changed (Durie et al. 1989; Hudson and Russel, 2009).

In 1988, the Royal Commission on Social Policy made a gesture to establish interpretations of the Treaty that would
be applicable in modern Aotearoa-NZ society (Durie, 1994; Hudson and Russel, 2009). These interpretations have been further





refined (Waitangi Tribunal, 2016) and are known as the Principles of the Treaty. The Principles of the Treaty, developed by the Waitangi Tribunal, intend to ensure that interactions between Māori and Crown entities—including research interactions—are ethical and within the stipulations of the Treaty. Select resource-specific principles (Brierley et al., 2018) indicate that the right to establish the spiritual and cultural significance of certain landscape features and resources remains with tangata whenua (Harmsworth et al., 2016).

### 220 3.2.2 The Treaty in practice

The Principles of the Treaty mandates that scientific investigations must consider the applicability and appropriateness of including Māori in research projects. Moreover, the Principles of the Treaty appears to reflect the te reo Māori version of the Treaty, which refers to depths of knowledge and implicitly includes science within the construct of mātauranga. Several research projects conducted within Aotearoa-NZ over the past few years exemplify the Treaty in practice. Harmsworth et al.
(2016) outline Aotearoa-NZ legislative frameworks that apply the Treaty of Waitangi to modern research endeavours. Here, we discuss two major advances in culturally responsive legislation.

### 3.2.2.1 Te Manahuna Aoraki Project

The Department of Conservation (DOC) is Aotearoa-NZ's government agency for conservation of national heritage, both natural and historic. DOC has a strict consultation procedure for engaging with iwi, hapū and whānau. The consultation process
is meant to uphold DOC's status as a Treaty partner, and employs the principles of partnership, protection, redress and reconciliation, and informed decision making (Department of Conservation, n.d.).

A modern and on-going example of the DOC consultation process with iwi is through Te Manahuna Aoraki Project. The players in this project are DOC, the NEXT foundation, Te Rūnanga o Waihao, Te Rūnanga o Moeraki, Te Rūnanga o Arowhenua, and others (Te Manahuna Aoraki Project, 2018). The iwi are official partners, which elevates their status from
stakeholder to decision-maker (Jo McLean, in Booth et al., 2019). The consultation process is not easy, however, as not all players will have the same priorities. For mana whenua, spiritual values of the Te Manahuna, the Mackenzie basin, are held as a priority to be conserved, which may be challenging to communicate to their partners (Jo McLean, in Booth et al., 2019). However, both Pākehā and Māori parties recognise Te Manahuna as a place of vitality, which can enable mutual respect for partners and the landscape.

The consultation process is still in its early stages (Jo McLean, in Booth et al., 2019), but, the purpose is to make a transformational shift in the way that organisations come together to deliver outcomes (Suzette van Aswegen, in Booth et al., 2019). Though this project is for conservation and management, there are many lessons that can be transferred to geomorphic research. Early consultation, legitimate partnership with iwi, sustained discussions and fair consideration of all key players' views are essential for a successful project that involves Māori and non-Māori researchers.



### 3.2.2.2 Te Awa Tupua

In 2017, the Whanganui River on the North Island of Aotearoa-NZ gained the status of legal personhood (Brierley et al., 2018). Te Awa Tupua was declared as "an indivisible and living whole from the mountains to the sea, incorporating the Whanganui River and all of its physical and metaphysical elements" (Te Awa Tupua [Whanganui River Claims Settlement] Act, section 13(b), 2017). Granting a river system personhood rights reflects a Māori approach to river system interaction and understanding. After Te Awa Tupua was legally recognised, Brierley et al. (2018) defined the geomorphic implications of the act. The authors posited that the river now has the following geomorphic "rights" (Brierley et al., 2018, p.4):

1. "A right to flowing water, and associated spatial and temporal variability in hydrologic and hydraulic regime.

2. A right to convey sediment, adjusting the balance of erosional and depositional processes in any given reach, and how these reaches fit together at the catchment scale, as materials are transported from "source to sink."

3. A right to be diverse, reflecting geographic and historical controls upon the inherent geodiversity (i.e., heterogeneity and/or homogeneity) of a river reach.

4. A right to adjust, shaped by mutual interactions between flow, sediment, riparian vegetation, wood, ecosystem engineers, and groundwater that set the dynamic habitat mosaic of river systems.

5. A right to evolve, set by responses to disturbance events and changes to boundary conditions that influence the trajectory of geomorphic adjustment of a river.

6. A right to operate at the catchment scale, as connectivity relations determine how changes to one part of a river system impact elsewhere in the catchment, and at the coastal interface, over what timeframe.

7. A right to be healthy, operating as a living river that maintains its integrity, vigour, and vitality, maximizing its resilience to impacts of disturbance."

Essentially, Brierley et al. (2018) state that the river has the right to be a river, the right to flow freely and transport sediment from the mountains to the sea. The river has a right to be a living system (Salmond et al., 2019). Brierley et al. (2018) argue that Te Awa Tupua was a milestone achievement in river management and geomorphologic research because river scientists created research questions that reflected both societal and environmental values. This act has implications for future legal interactions concerning mātauranga Māori, Māori worldview, science, landscape research priorities and conservation efforts (Ruru, 2018; Geddis and Ruru, 2019).



### 3.3 Woven spaces—the interface of mātauranga Māori and science

#### 3.3.1 The relationship between mātauranga and science

Like with many Indigenous knowledge systems, mātauranga Māori has historically been 'systematically dismissed and erased… as being worthless' (Waitangi Tribunal 1999, *as cited in* Broughton et al., 2015). However, when expressed in a way
to which Western scientists can relate, it is clear that pre-European Māori generated some of their knowledge in ways consistent with the scientific method (Cunningham, 2000; Hikuroa, 2017). Over the past decade, select Māori researchers in the physical sciences (e.g., King et al., 2007; Tipa, 2009; Harmsworth et al., 2016; Hikuroa, 2017; Hikuroa et al., 2018; Paul-Burke et al., 2018) have made strides for advancing mātauranga alongside Western science. These researchers have promoted the mana of mātauranga and advocated for its place in national research through their own research endeavours. As a result of the efforts
of these researchers, as well as others in different fields (e.g., Durie, 2004; Smith, 2012; Macfarlane et al., 2015), the Aotearoa-NZ government now requires an acknowledgement and consideration of research relevance to Māori in many major grant and funding applications such as the Hīkina Whakatutuki Ministry of Business Innovation and Employment's Endeavour Fund, National Science Challenges and Te Punaha Hihiko: Vision Mātauranga Capability Fund (Hīkina Whakatutuki Ministry of Business, Innovation and Employment, n.d.). Notably, in 2011, the Vision Mātauranga policy statement was incorporated into
the Statements of Core Purpose of Crown Research Institutes (CRIs), which requires CRIs to enable the potential for innovative research with Māori.

Perhaps the major difference between Indigenous knowledge (here, mātauranga) and science are perceptions of objectivity. Within a Māori worldview, humans sit in the heart of natural systems, along with all other components (Hikuroa, 2017). In a scientific worldview, objectivity is essential for making unbiased observations to test hypotheses (Moller, 2009;
Crawford, 2009). Facts and values are separated (Hikuroa, 2017). But in the Māori worldview, knowledge is informed by values and values are informed by knowledge. There is no separation between values and facts. Understanding this interplay, and acknowledging and respecting the potential values that Indigenous knowledge can bring to science, is paramount for successful research at the interface (Pingram et al., 2019). Rather than contesting relative validities, Durie (2004) and Peet (2006) demonstrate that work at the interface can be a space for inventiveness and inspiration. Nevertheless, Mercier (2007)
cautions that focusing on the difference can create a position in which science and Indigenous knowledge can potentially clash.

#### 3.3.1.1 Knowledge versus values

Māori values can, in part, be understood as the means by which Māori make sense of and understand their environment (Marsden, 1988 *as cited in* Harmsworth and Awatere, 2013). Examples of these values include tikanga, whakapapa, tino rangatiratanga, mana whenua, whānaungatanga, kaitiakitanga, manaakitanga, whakakotahitanga, arohatanga and wairuatanga
(Table 1; Harmsworth and Awatere, 2013). Māori values directly inform mātauranga Māori (Hikuroa, 2017), and mātauranga informs Māori values (Harmsworth and Awatere, 2013). This reciprocity between values and knowledge may be considered a major difference between mātauranga generation and Western scientific knowledge generation. Western science is informed





by truth and evidence, whereas mātauranga is informed by fact and value (Hikuroa, 2017). However, because mātauranga Māori and Māori values are both traditional and contemporary, Māori perspectives have the potential to contribute to
innovative research approaches in which knowledge is co-created considering both Māori and Western values.

### 3.3.2 Mutual research needs and benefits

As discussed earlier, The Aotearoa-NZ government outlined a goal for research at the interface of Western science and mātauranga Māori in their 2007 Vision Mātauranga statement: *To unlock the innovation potential of Māori knowledge, resources and people to assist New Zealanders to create a better future* (Ministry of Research, Science and Technology, 2007).
One of the four Vision Mātauranga research themes is *Taiao*: Achieving Environmental Sustainability through Iwi and Hapū Relationships with Land and Sea. This theme explores iwi and hapū relationships with land and seascapes, and encourages Māori involvement in research relating to the sustainability of these environments. This official document is a tool for researchers considering different projects and their applicability to Māori. Though Vision Mātauranga does not explicitly outline how to conduct research at the interface (Macfarlane and Macfarlane, 2018), it establishes the context for bicultural
approaches to research.

Iwi management plans (IMPs) and iwi environmental management plans (IEMPs) are official documents that can be used to define iwi-identified research needs. Extensive work has been completed to highlight the utility of IMPs as guides and frameworks for engagement with Māori (Saunders, 2017). IMPs provide clear, official documentation of iwi values and interests that can be considered in research (Waikato Regional Council, 2019). Many IMPs and IEMPs discuss iwi goals for
minimum river flows and flood hazards (e.g., Tipa et al., 2014), which are specifically relevant to geomorphologists. Some plans have sections with specific goals for rivers (e.g., Waikato-Tainui Te Kauhanganui Incorporated, 2013) or catchments (e.g., Te Rūnanga o Kaikōura et al., 2005; Jolly and Ngā Papatipu Rūnanga Working Group, 2013). Most IMPs are focused on improving the mauri of tribal landscapes.

IMPs provide the opportunity for mātauranga Māori to be included in planning and research projects as a knowledge
system parallel to Western science (Saunders, 2017). In addition to outlining key values and interests, IMPs provide specific guidance to researchers and planners on how each iwi proposes consultation and engagement activities might proceed.

### 3.3.3 Potential challenges and risks of conducting research at the interface

It is essential to note that there may be circumstances when it is inappropriate to draw upon both Western science and mātauranga Māori. Mika and Stewart (2017) in fact advocate that perhaps it is better to maintain a binary research sphere all
together, wherein Western and Māori approaches are kept separate. There may be situations when one explanation (i.e. Indigenous) for an event does not align with another explanation (i.e. Western). For example, research concerning oral histories of meteor impact craters in Australia indicate that it is possible that events recorded in oral histories cannot be correlated with physical scientific evidence (e.g., Hamacher and Norris, 2010) or that some landscapes do not have associated oral histories (e.g., Hamacher and Goldsmith, 2013). In cases such as these, it is essential to maintain mutual respect by not using one method



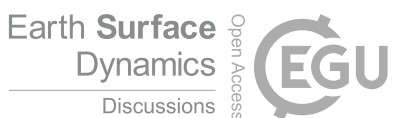

to prove the other method wrong (Durie, 2004). Accordingly, science and mātauranga should not be used to prove each other wrong (Hikuroa et al., 2011; Hikuroa, 2017). It becomes the researcher's responsibility to determine which method provides the stronger supporting evidence, but not to the exclusion of identifying and discussing inconsistencies and discrepancies. If done appropriately, it is possible for the two approaches to strengthen one another and provide better outcomes for all involved (Durie, 2007). These could be opportunities to explore the richness and contingency of oral traditions separate from

scientifically-determined landscape events. Equally, oral traditions could be the only record of something that was perishable in the geomorphological/geological record. In cases where Indigenous knowledge does align with scientific findings, the supporting evidence is purely stronger.

        A Māori worldview accepts that there can be more than one explanation for an event or landform. The concept of contested knowledges and opposing viewpoints between Indigenous communities was an accepted part of life (Smith, 1999).

This led to creating an environment of tolerance, mutual respect and reciprocity between Indigenous communities. Multiple ideas or explanations for an event is also common in the field of geomorphology, where landscape formation can be explored through multiple working hypotheses via the principle of equifinality. While conducting research at the interface poses many challenges, it reveals similarities such as these and presents opportunities to generate corroborative evidence for events and landforms.

**4. Frameworks and models for incorporating mātauranga Māori alongside geomorphic research**

Extensive work has been done by Māori researchers to develop frameworks and models for including Māori knowledge, values and tikanga in research. Smith (1992) established and promoted ways for non-Māori researchers to engage with Māori and maintain a standard of cultural responsibility. Smith (2012) later described kaupapa Māori research, or research by Māori for Māori, and detailed appropriate ways for Māori to lead their own research aspirations. The models proposed by Smith (1992,

2012) can be thought of as methodologies, or guiding principles according to which researchers define research questions, select tools and approaches to address questions, and plan analyses (Harding, 1987; Smith, 2012).

        Harmsworth et al. (2016) discuss models for integrating Māori values into environmental research. These models can be thought of as methods, actions or procedures by which a researcher addresses the core research questions and collects data (Smith, 2012). Similarly, Rainforth and Harmsworth (2019) provide a detailed and extensive review of tools, frameworks and

methods that have been developed to include iwi and hapū values into freshwater management. Like international research that aims to weave Indigenous knowledge and values with Western scientific techniques, most of the frameworks reviewed by Rainforth and Harmsworth (2019) reflect a strong effort to include mātauranga Māori in ecological and environmental assessments, but a lack of studies that weave mātauranga Māori with geomorphic research. The models that Harmsworth et al. (2016) and Rainforth and Harmsworth (2019) discuss are Indigenous approaches to research. These approaches include

Indigenous values and cultural protocols (Smith, 2012). Indigenous approaches to research like these are commonly structured



as models or decision support tools (Morgan, 2006) that empower Indigenous values alongside Western practices (Hikuroa et al., 2018).

This section introduces three theoretical frameworks for including or considering mātauranga Māori in geomorphic research. The frameworks discussed here have been previously analysed in the health and education contexts (e.g., Macfarlane et al., 2015; Macfarlane and Macfarlane, 2018). We discuss how each theoretical framework could be transferrable to geomorphic research. Keeping in mind that mātauranga and values cannot always be separated, we then introduce three models for including Māori values within science conducted according to Western practices and highlight how each model could be transferred to geomorphology. A critical assessment of the frameworks and models is provided in section 5, and a discussion of how these frameworks and models can be applied outside of Aotearoa-NZ is provided in section 6.

### 4.1 Theoretical frameworks for including mātauranga Māori in geomorphic research

The following frameworks are theoretical guiding frameworks for including mātauranga Māori in research projects. The three reviewed here have been discussed by Macfarlane and Macfarlane (2018), but here we also discuss their applicability to geomorphology. Although these frameworks were developed and promoted by researchers seeking better outcomes in the health and education sphere, we do not believe they are necessarily only applicable to those spaces.

### 4.1.1 He Poutama Whakamana

The He Poutama Whakamana framework draws directly from principles that reflect the intent of the Ministry of Research, Science and Technology's 2007 Vision Mātauranga policy (Macfarlane and Macfarlane, 2018). He Poutama Whakamana alludes to mirror-imaged panels—Poutama Tukutuku—that are typically present in traditional Māori meeting houses. These Poutama Tukutuku represent a journey of growth and learning in order to metaphorically climb up to where knowledge and understanding are achieved.

Macfarlane and Macfarlane (2018) propose that this is a good framework for including Māori phenomena into research. There are three main steps of the framework: mōhiotanga (acknowledgement, respect), mātauranga (knowledge, understanding) and māramatanga (integration, application) (Fig. 3). Each of the three steps individually and uniquely addresses four principles from Vision Mātauranga: kaitiakitanga, mātauranga, tikanga and rangatiratanga. These four principles reappear in each of the three steps, with different implications in each iteration (Fig. 3a). He Poutama Whakamana follows a kaupapa Māori research approach. Kaupapa Māori, described in depth by Smith (2012), can be understood as research that is "culturally safe" and that takes place within a Māori worldview (Irwin, 1994 *as cited in* Smith, 2012). There is space for non-Indigenous researchers within a kaupapa Māori approach (Bishop, 1994 *as cited in* Smith, 2012).

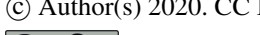



Poutama Tukutuku, as previously mentioned, are mirror-imaged panels, which presents a metaphorical space for the

scientific method to operate alongside the kaupapa Māori theme (Fig. 3b). There is no hindrance to the scientific method (Fig. 3b), but it requires additional "check-ins" throughout the process to make sure that the Vision Mātauranga principles are being reflected in both themes. Adequately addressing Vision Mātauranga principles in both themes has the potential to ultimately produce co-created knowledge (Fig. 3c). This approach also has the potential to emphasize the differences between the two approaches (e.g., Mercier, 2007), which may in its own right lead to better understandings and outcomes. He Poutama

Whakamana is suitable for geomorphic research because it is open-ended and not specialised for any one field of research. It

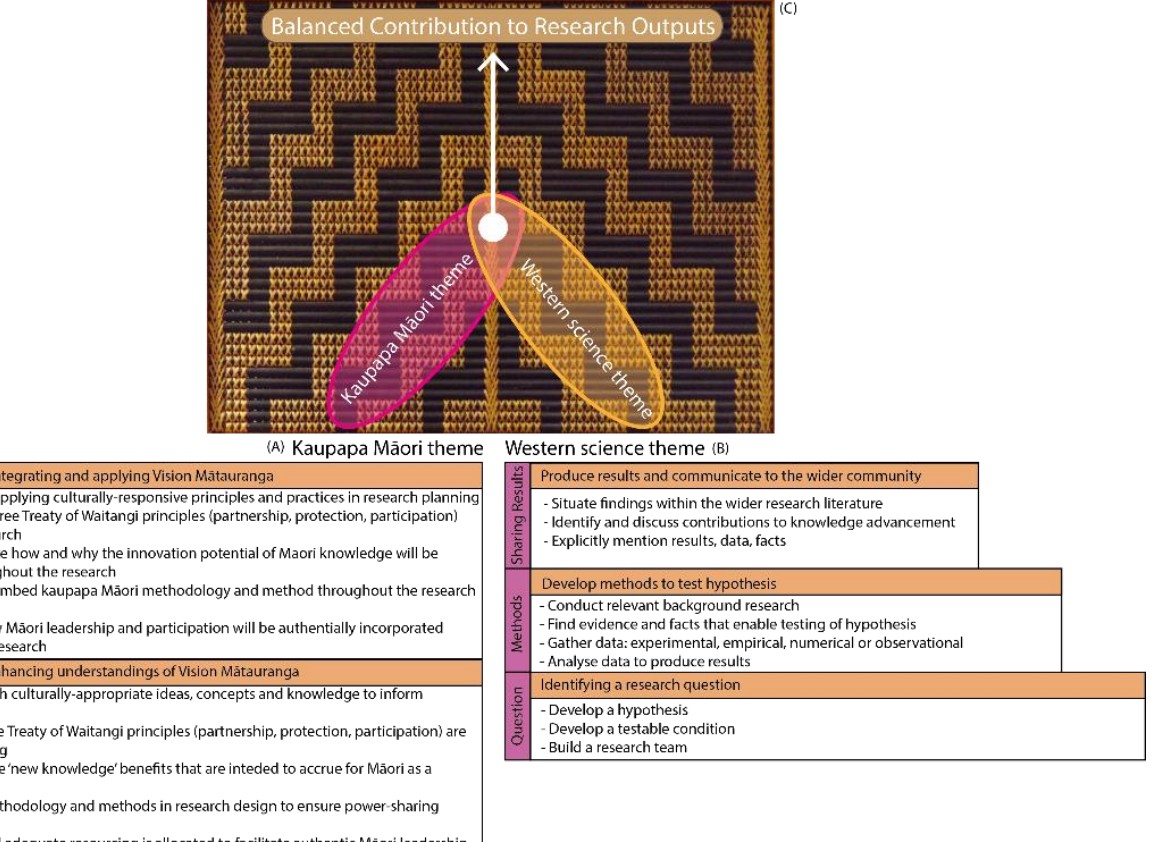

**Figure 3. A) The He Poutama Whakamana framework (Macfarlane and Macfarlane, 2018) mirrored by B) the scientific method theory on the opposite side of the Poutama Tukutuku. C) represents upwards growth towards co-creation of knowledge.**





welcomes research conducted under the guidance of the Treaty of Waitangi, and as long as each step of the framework is addressed, the research will potentially be culturally responsible and safe.

### 4.1.2 IBRLA

The IBRLA (initiation, benefits, representation, legitimation, accountability) framework is also an open-ended research
framework that aims to ensure that Māori thinking and voice are included in research involving Māori (Bishop, 1996; Macfarlane and Macfarlane, 2018). It features a series of accountability questions within each component of the framework (Table 3). These questions are meant to guide researchers and help ensure that Māori knowledge is being included throughout the research project. These questions, such as "How did Māori participate in the conceptualisation and initiation process?" or "How will Māori thinking and knowledge be represented at all research phases?" hold researchers responsible for ensuring
that Māori involvement and contribution is not only included but prioritised in the research. Again, there is no hindrance to

Table 3. The IBRLA framework. *Adapted from* Macfarlane and Macfarlane, 2018.

| | Principle | Accountability Questions |
|---|---|---|
| I | Initiation | • Who conceptualised and initiated this research project?<br>• How did Māori participate in the conceptualisation and initiation process?<br>• How was the agreement to proceed with the research achieved? |
| B | Benefits | • How will the research (process and outcomes) accrue benefits for Māori?<br>• How has information been shared with Māori about the intended benefits?<br>• How will these benefits be determined and measured—and by whom? |
| R | Representation | • Whose ideas will be represented in the methodology, design and approach?<br>• How will Māori thinking and knowledge be represented at all research phases?<br>• How will this be monitored so that ongoing agreement/partnership is maintained? |
| L | Legitimation | • Who will legitimate the analysis and interpretation of information/research data?<br>• How will Māori understandings be legitimately represented?<br>• How will this be structured so that research fidelity is achieved/protected? |
| A | Accountability | • Who is accountable to whom—and in what ways?<br>• How will on-going and mutual accountability be built into the research process?<br>• How will this be monitored and evaluated to ensure safety for all stakeholders? |





using the scientific method within this research framework, but the questions help ensure that mātauranga Māori is respected and upheld throughout the research process.

Just as the scientific method often encourages revisiting hypotheses, the IBRLA framework encourages researcher reflection during the concept design stage (similar to hypothesis formation and method development) through to the end of the
research. The intent of IBRLA is to produce collaborative research stories (Bishop, 1996). This framework can provide a sense of researcher security when including Māori knowledge, while maintaining the integrity of the scientific method.

### 4.1.3 He Awa Whiria

The He Awa Whiria methodology explicitly recognises the benefits of both the Western science paradigm and kaupapa Māori principles. A project designed under the He Awa Whiria methodology has two streams, representing the aforementioned
approaches to research (Fig. 4). Throughout the duration of the project, parts of one stream may widen or increase in strength while the other narrows and assumes a lighter role. At other times, the weaker stream may gain momentum, shifting the balance of the overall research project in the other direction. The streams converge and diverge throughout the project, and the moments of convergence are times of learning. Though the streams may wane or grow in strength, change directions, or even die out in places (as the channels in a braided river do), both streams have the same objective, which is to provide balanced contributions
to research outcomes. It is the researcher's role to manage how and when the two streams must converge, and when it is appropriate for them to diverge. Ultimately, when research conclusions are drawn, the claims must be supported by both streams.

The He Awa Whiria methodology allows for flexibility within a research project. It enables Western science to stay true to the scientific method. It also provides grounds for mātauranga Māori input to focus the Western science analysis.
Wilkinson and Macfarlane (in press) demonstrate that the He Awa Whiria method can be applied to geomorphic studies by allowing the knowledge two streams to operate both independently and collaboratively. The He Awa Whiria methodology supports a culturally responsible and responsive approach to research and allows for variable approaches to research depending on the specific topic. Methodological adaptability is essential for conducting research with Māori, because different Māori groups will have different values, priorities and interests when it comes to pursuing research questions.

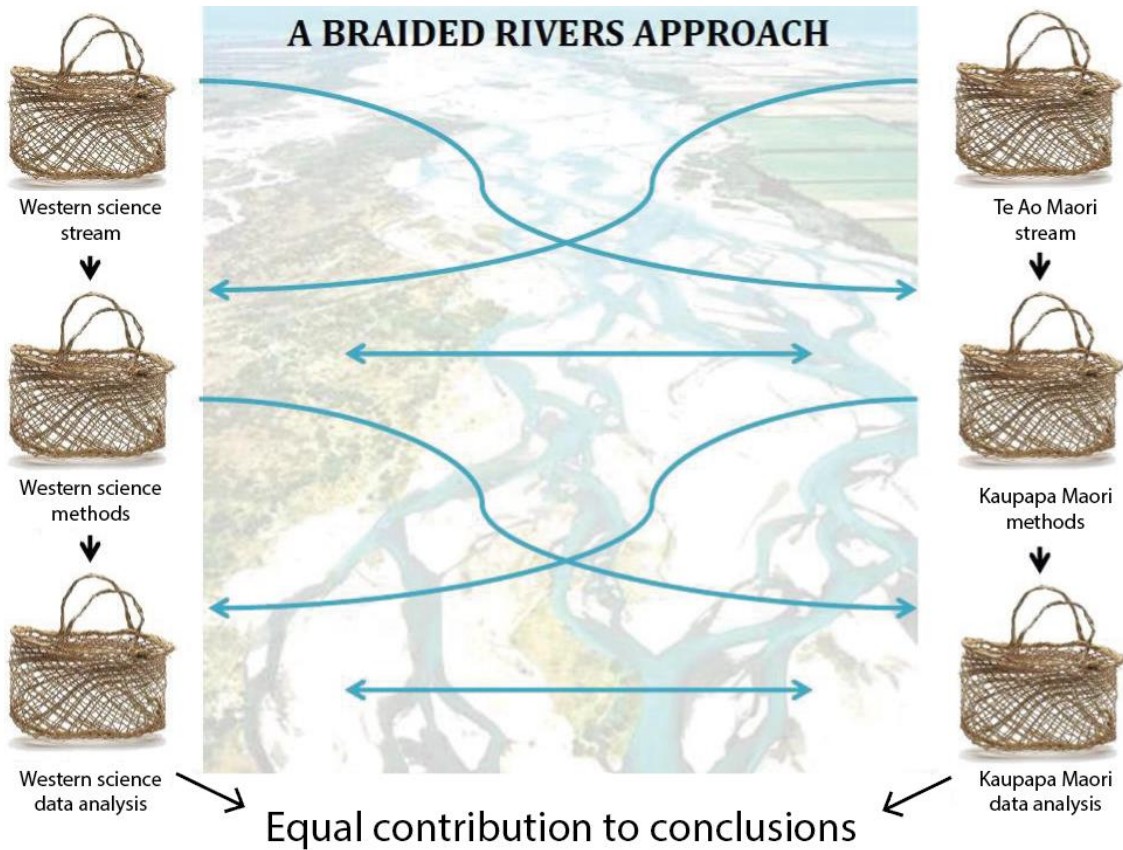

**Figure 4. The He Awa Whiria framework.** *Modified from* **Macfarlane et al., 2015.**

**4.2 Models for including Māori values in geomorphic research**

The following models are step-by-step processes for explicitly including Māori values into scientific research. Many of these models were originally designed to assist with environmental decision-making and management. These models can be incorporated into the knowledge-inclusion frameworks above, creating research projects with nested methods and methodologies.



### 4.2.1 Mauri model


The Mauri Model was developed as a tool for creating sustainable solutions for environmental decision-making in Aotearoa-NZ (Morgan, 2006; Faaui et al., 2017). It is grounded in the Māori concept of mauri, which can be best understood as an ethereal bond that links all elements of the natural world, the binding force between the physical and the metaphysical, the life-supporting capacity of soil and water. The Mauri Model is a decision-making framework and provides a template for the

explicit inclusion of Indigenous values with Western knowledge (Morgan, 2006; Hikuroa et al., 2011). The aim of the Mauri Model is to define the degree of sustainability of proposed projects or activities by assessing the impact of an action on the mauri of an area (Hikuroa et al., 2011). The model considers a wide range of environmental, cultural, social, and economic indicators for use in analysis. Each indicator receives a value from a scale of -2 to +2, with -2 being mauri noho/mate (denigrated), -1 being mauri heke (diminishing), 0 being maintaining, 1 being mauri piki (enhancing) and 2 being mauri tu/ora

(restored) (Fig. 5). The Mauri Model can work independently of science but is most effective when science is integrated into the analysis.

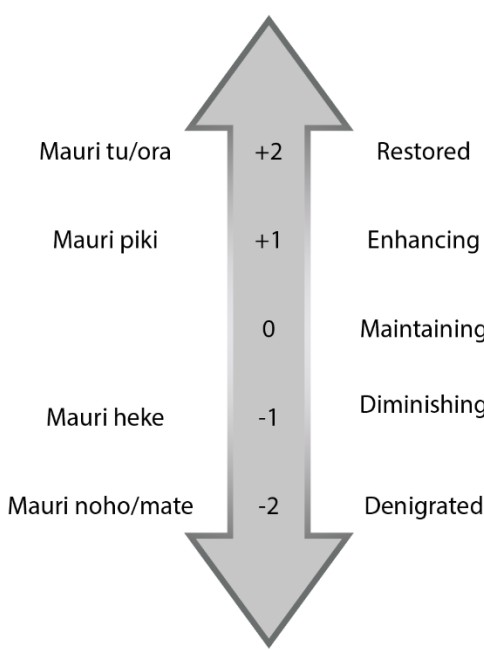

**Figure 5. "Mauri Meter" (Morgan, 2006); Infographic of the valuing system in the Mauri Model.** *Modified from* **Hikuroa et al., 2011.**





To complete the assessment, each indicator is listed in a table and the integer values of each indicator are summed to determine the impact on the area of interest's mauri. This model provides a semi-quantitative assessment of the impact on mauri. This method could be appealing to researchers or project managers working in bicultural spaces because it combines

stakeholder interests with Indigenous values in a semi-quantitative framework (Morgan and Fa'aui, 2018).

Morgan (2006) originally developed the Mauri Model to create a tool that could be utilised to include Māori input on water management issues in Aotearoa-NZ. It has since been used nationally and internationally to conduct environmental assessments in post-disaster maritime settings (Faaui et al., 2017), in geothermal development areas (Hikuroa et al., 2010) in areas of high anthropogenic modification (Hikuroa et al., 2018) and in dam impact studies (Morgan et al., 2012). Hikuroa et

al. (2018) provide an extensive list of studies that have utilised the Mauri Model both within and beyond Aotearoa-NZ.

#### 4.2.1.1 Transferability to geomorphology

Although the Mauri Model was designed as an assessment for the impact of human activities on an area, the ideas of denigrated and diminishing, or enhancing and restored landscapes can be transferred to geomorphology research. For example, in 2016 an $M_w$ 7.8 earthquake struck the Kaikōura region on the South Island of Aotearoa-NZ (Hamling et al., 2017). The

earthquake caused over 20,000 landslides that delivered mass amounts of sediment to river catchments (Massey et al., 2018). Fine sediment has been carried to the sea, and has smothered and suffocated tidal to intertidal shallow-marine ecosystems (Schiel et al., 2019). Coupled high sedimentation and coastal uplift has caused the biogeomorphology of the region to change dramatically following the earthquake (Schiel et al., 2019). The ongoing stability of marine species has the potential to indicate sedimentation rates and the effect that the geomorphology of the area has on marine populations. In a study of Te Awa o te

Atua (Tarawera River), Hikuroa et al. (2018) show that sedimentation is a contributing factor in the Mauri Model assessment. Therefore, we hypothesise that the Mauri Model could be applied to research investigating the effects of a natural geologic and geomorphic event (rather than a specifically human-induced act) on an ecosystem or landscape.

#### 4.2.2 Cultural Flow Preference Study

The Cultural Flow Preference Study (CFPS) model was developed as a tool for Māori to assess their ability to engage in

cultural practices within catchments at certain river flow levels (Tipa and Nelson, 2012) and to engage with freshwater resource management decisions (Crow et al., 2018). The CFPS model falls under the process of Cultural Opportunity Mapping, Assessments and Responses (COMAR), which are integrated processes that empower mana whenua to engage in freshwater studies and management (Tipa and Nelson, 2008). A CFPS can be used to help Māori engage with research projects and management plans for freshwater environments (Crow et al., 2018).





As the CFPS is heavily site-specific, it demonstrates the benefits that hapū and iwi knowledge and values can bring to modern river management and scientific endeavours (Tipa, 2009b; Crow et al., 2018). The CFPS methodology accounts for variations in cultural values between whānau, hapū and iwi by providing a framework that can be transferred and applied for different studies. The first step of a CFPS is to identify the tangata whenua team, who act as the leading experts for a specific river or area, and determine the cultural values held by that team. After the tangata whenua team has been formed, a series of

steps are followed in order to create a CFPS (Fig. 6). The ultimate aim of a CFPS is to link cultural values to variations in river flow, and to determine how cultural values change depending on the flow of the river.

| |
|---|
| Step 1: Identify cultural values associated with the river, especially site-specific values, using the tangata whenua team |
| Step 2: Identify the flow dependencies of values and attributes that need to be included in the CFPS |
| Step 3: Apply the appropriate field assessment for all cultural attributes |
| Step 4: Relate to other flow assessment methods e.g., ecological, economic, etc. |
| Step 5: Apply cultural methods |
| Step 6: Quantify relationship between flow and cultural values |
| Step 7: Analyse data to recommend flow regimes that reflect identified values, beliefs, or practices and create a goal for protection |
| Step 8: Scientific monitoring informed by cultural values—does the flow provide the intended outcome? |
| Step 9: Report results and apply to management |

**Figure 6. Steps to complete a CFPS.** *Modified from* **Crow et al., 2018.**

**4.2.2.1 Transferability to geomorphology**

CFPSs are applicable to geomorphic studies, specifically fluvial geomorphology. Tipa and Nelson (2012) demonstrate the utility of applying a CFPS in a study concerning the Kakaunui Catchment, South Island Aotearoa-NZ. During this process,

they followed the CFPS method to: 1) identify their tangata whenua team (Te Rūnanga o Moeraki); 2) have the tangata whenua team define their cultural association with the river; 3) conduct a participatory mapping exercise to identify how the local iwi valued the river, what hydrological characteristics the local iwi believed to be essential to protect those values, how current and historic hydrologic and geomorphic characteristics differ, and how the current flow rates affect cultural values and uses;





4) identify and analyse tangata whenua-identified flow issues; and 5) calculate minimum flows that would satisfy cultural flow
preferences. Through this method, Tipa and Nelson (2012) concluded that the current minimum flow in the Kakaunui
Catchment (250 L/s) is likely too low to maintain Te Rūnanga o Moeraki's values within the catchment. This study allowed
geomorphic, hydraulic, ecologic and cultural values to be considered in tandem. Identified tangata whenua values helped drive
the research intentions and resulted in an outcome that could have application in future management of the Kakaunui
catchment.

**4.2.3 Sustainability Assessment Method**

The Sustainability Assessment Method (SAM) is another environmental monitoring tool for freshwater catchments that can
be used to include Māori values alongside more traditional monitoring assessments (Tipa, 2009b). The SAM explicitly includes
social, cultural, economic and environmental values. This multi-dimensional assessment is laid-out in a step-by-step guideline
that enables researchers to document cultural values and associations with river catchments alongside scientific monitoring
techniques (Fig. 7). The cultural dimension of this model focuses strongly on water quality and typically uses mahinga kai as
an indicator of the health of waterways.

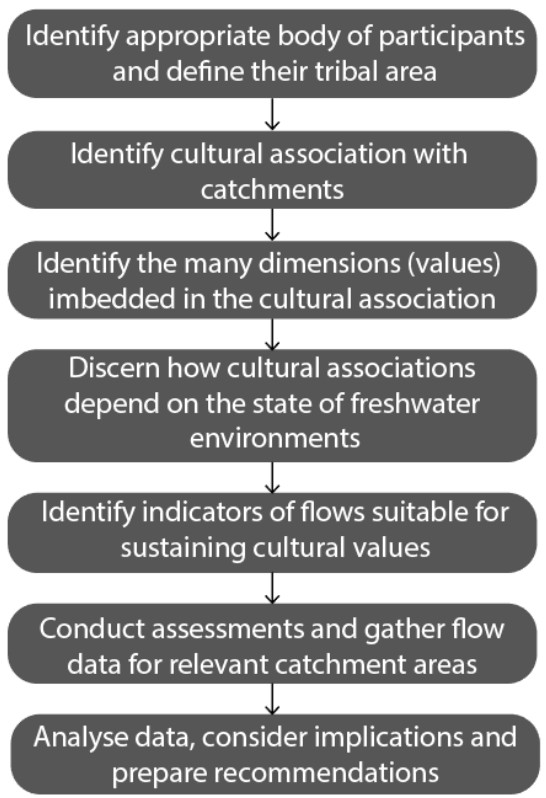

**Figure 7. Steps in a SAM.** *Modified from Tipa, 2009.*



The SAM follows a similar trajectory as other research frameworks involving Māori. The first step is to identify the appropriate group of tangata whenua participants, document their cultural relationships with a catchment, and ultimately determine baseline water quality and quantity standards compliant with Māori preferences. Tipa (2009) suggests that this
model can be used as an alternative to strictly Western-style freshwater assessments, but it is possible that this model could be included alongside a Western-style analysis to build a more comprehensive assessment. Māori involvement is required, and the final step—analyse data, evaluate implications, prepare strategies and recommendations—should include both Māori and Western interpretations of the results.

### 4.2.3.1 Transferability to geomorphology

In 2005, the SAM was adapted for use in an assessment of New Zealand river catchments from Māori perspectives (Tipa 2007 as cited in Tipa, 2009). Lists of Māori values, beliefs and practices associated with three river catchments in the South Island of Aotearoa-NZ were accumulated from analyses of contemporary writings and historical accounts. From the lists, tables were constructed to describe all concepts that described a value, belief or practice that surfaced from the initial analyses. Using the SAM allowed Māori concepts to be organised in a way that each element could be examined separately, in the context of each
individual river. Beauty, mahinga kai, water quality and Ki Uta Ki Tai (from the mountains to the sea) are a few examples of the many identified by tangata whenua as important values within these catchments (Tipa, 2009). The result of this exercise was to show that the SAM could give resource managers the opportunity to consider cultural values alongside westernised resource management priorities. The SAM promotes a tool for policy makers that incorporates a place-based approach, allowing for more specialised outcomes.
525       As indicated by Tipa (2009), it would be possible to also include geomorphic values alongside a SAM analysis. Recalling Brierly et al.'s (2018) geomorphic rights of the river, the SAM would enable river geomorphologists and managers to apply equitable consideration to both Indigenous and scientific values in river management strategies or research projects. The requirement would be that the team includes members who are experts—either individually or collectively—in both mātauranga and scientific techniques. That way, geomorphic values can be considered alongside the cultural values proposed
by the tangata whenua team. This sort of approach could yield better river management outcomes for both Māori and non-Māori.

### 5. Critical assessment of existing frameworks in different conditions

These frameworks and models do, ideally, require Māori guidance and Māori participation on the research team. In many cases, it may be appropriate to select a theoretical framework to guide research methodologies and, if appropriate, apply a
value-based model within the research framework to act as a guide for the project's methods. This section provides an analysis of the presented theoretical knowledge frameworks and the value-based models, offers recommendations for geomorphic subdisciplines, and provides information about how researchers can identify research questions using Māori priorities.



## 5.1 Knowledge versus values revisited

The theoretical research frameworks (e.g. He Poutama Whakamana, IBRLA, He Awa Whiria) are systems to weave Māori
worldview and knowledge into or alongside many research disciplines, including geomorphology. These frameworks can be
thought of as methodologies, or philosophical processes according to which research is conducted (Harding, 1987; Smith,
2012). These frameworks support and promote Māori knowledge and ensure that mātauranga is prioritised throughout the
research process.

Explicitly including Māori values into research can be achieved by nesting value-based models (e.g., Mauri Model,
CFPS, SAM) within the aforementioned research frameworks. Value-based models can be thought of as methods, or specific
steps, actions or procedures that researchers can follow to answer core research questions and collect data (Smith, 2012).
Indigenous methods include values, customs, protocols and existing knowledge. When value-based models are nested in
Māori-focused theoretical research frameworks, the interconnectedness between values and knowledge becomes apparent.

There may be times when it is easier for non-Māori researchers to include Māori knowledge by way of Māori values.
Value-based models are an adequate way to follow step-by-step processes (similar to research processes produced according
to the scientific method) that address Māori ways of knowing and living. However, regardless of how Māori knowledge is
included, it is ideal to have a Māori researcher on the project leadership team and to have initial consultations with university
or research organisation iwi engagement and support teams. Early Māori involvement is key for identifying appropriate tangata
whenua groups, who can aid in new knowledge generation (Broughton et al., 2015).

## 5.2 Framework recommendations for subdisciplines

The three theoretical frameworks presented here all have the ability to be transferred to geomorphic research. They do not
preclude researchers from using the scientific method to produce knowledge, but they do require that researchers use a kaupapa
Māori approach to also co-create knowledge with Māori.

It may be best to select theoretical frameworks based on the distribution of Māori to non-Māori researchers involved
in the research project. The He Poutama Whakamana and IBRLA models may be most appropriate for research teams where
the research team has a majority of non-Māori leadership, because they are not strictly expressed through a Māori worldview.
These policies are rooted in Aotearoa-NZ governmental policy—the Treaty of Waitangi and Vision Mātauranga—and provide
explicit checks and balances for researchers. Researcher reflection is a major element of these frameworks. The He Awa Whiria
model may be suitable for research teams comprising any ratio of Māori to non-Māori leaders. Because the two research
streams converge, diverge, and act dynamically for the entirety of the project (Macfarlane et al., 2015; Macfarlane and
Macfarlane, 2018), it may be possible for one stream to have a larger sub-leadership team than the other. This framework
specifically allows for mātauranga Māori to focus the Western science stream. This balance will vary from project to project,
but a project will successfully adhere to this framework as long as the interactions that do occur between the two streams foster
learning and new knowledge generation (Macfarlane and Macfarlane, 2018).



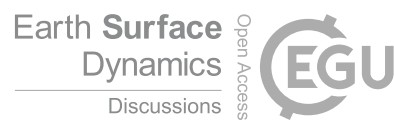

So what framework might a glaciologist, or a fluvial geomorphologist, or a pedologist, choose to ensure that they include Māori knowledge in their research? Selecting the right framework will stem from conversations with the appropriate iwi groups or Māori researchers early in the research process, and it will depend on the expertise of the research team. There is a common theme throughout frameworks and value-based models that the first step is to identify the appropriate group of Māori participants, or tangata whenua, to act as the leading experts for their tribal areas. Once these individuals are identified,
framework selection can happen cooperatively between scientists and tangata whenua. Each of the frameworks discussed here allow flexibility within the project and allow kaupapa Māori principles to excel alongside the scientific method. We therefore propose that framework selection must be done on a case-by-case basis, and the correct framework for any given research endeavour will be the one that suits all parties.

### 5.3 Guiding resources for initiating projects in Aotearoa-NZ

As previously discussed, many iwi and hapū in Aotearoa-NZ have published iwi management plans or iwi environmental management plans that can outline research priorities for scientists (Saunders, 2017). Many IMPs contain information specifically relevant to geomorphologists. For example, most IMPS discuss iwi goals for minimum river flows and mitigating flood hazards (e.g., Tipa et al., 2014). Other iwi environmental management plans have sections with specific goals for rivers (e.g., Waikato-Tainui Te Kauhanganui Incorporated, 2013) or catchments (e.g., Te Rūnanga o Kaikōura et al., 2005; Jolly and
Ngā Papatipu Rūnanga Working Group, 2013). Most IMPs focus on improving the mauri of tribal landscapes over which the authoring organisation (typically a rūnanga) possesses kaitiakitanga. Researchers can use information outlined in IMPs and IEMPs to identify the appropriate research leadership team and select the appropriate research framework.

Research funding guidelines for projects that aim to include mātauranga Māori alongside Western science can be found through Aotearoa-NZ's Ministry of Research, Science and Technology. Specifically, the Ministry of Business,
Innovation and Education (MBIE) operates Te Punaha Hihiko, the Vision Mātauranga Capability Fund, which provides guidelines for research projects in its application process (Ministry of Research, Science and Technology, 2007). The Marsden Fund, through the Royal Society of New Zealand, also provides for how proposals should include consideration of Māori involvement in research (Royal Society Te Apārangi, n.d.).

Many universities and research organisations have iwi engagement and support teams. These teams are an excellent
resource for gaining guidance on identifying the best tangata whenua team for research needs, and how to appropriately engage with that iwi or hapū. In Aotearoa-NZ, the universities and CRIs, in particular, have excellent resources for connecting researchers with Māori. We recommend early and, ideally, regular interaction with these resource groups.

### 6. Lessons for the international geomorphology community

Indigenous knowledge around the globe is a valid source of information because it has endured for generations, keeping
populations alive and securing their livelihoods. Moreover, Indigenous knowledge has been shown to be accurate and precise





(Hikuroa, 2017). In this section, we outline some direct benefits of including Indigenous knowledge in geomorphic research, discuss how the frameworks detailed in this review can be adapted for use outside of Aotearoa-NZ, and discuss how Indigenous knowledge and geomorphic research can and are working together to inform sustainability policy and legislation.

### 6.1 Direct benefits to geomorphology

A clear benefit to geomorphology is the temporal extension of observations of geomorphic events into pre-history. The 400-year volcanic record discussed by Swanson (2008) and the cycles of flood and channel avulsion evaluated by Hikuroa (2017) indicate that Indigenous knowledge can bolster scientifically-investigated geomorphic histories. King and Goff (2006) further demonstrated that Māori oral histories frequently discuss multiple geomorphic phenomena happening in tandem or as cascading events. Recognition of the interconnectedness of landscape processes is a common theme in many Indigenous

societies (Riggs, 2005) and this recognition has resulted in a way of life that responds to, interacts with and learns from concurrent or cascading suites of local landscape processes.

        Another key benefit of including Indigenous knowledge in geomorphic endeavours is the opportunity to co-create new approaches to research that build holistic and more complete understandings of landscape processes, with Indigenous knowledge and traditional narratives providing signposts for initiating and conducting geomorphic research. The dynamic and

adaptive nature of Indigenous knowledge generation (Berkes, 2009) has the potential to guide flexible research methods, which are well suited to the interdisciplinary field of geomorphology. Flexible research methods, in turn, have the potential to generate robust data collection with information from a variety of sources.

        A prime example of how flexible research methods incorporating Indigenous knowledge can provide significant contributions to geomorphic research is the New Zealand Palaeotsunami Database. The database aims to catalogue all tsunamis

that occurred prior to written historical records and uses three types of evidence to identify palaeotsunami events: sedimentary/artefactual ("primary"), geomorphic ("secondary") and anthropological/pūrākau ("cultural") (Goff, 2008; New Zealand Palaeotsunami Database, 2017). The cultural information allowed the database compilers to better constrain the age of palaeotsunami events by dating archaeological sites that were associated with the cultural information (Goff, 2008). A typical prehistoric Māori response to big waves was to abandon coastal settlements and move to higher elevations (Goff and

Chagué-Goff, 2015). Cultural knowledge of the locations of abandoned sites allowed researchers to conduct archaeological investigations and date the time at which such sites had been occupied, thus providing a well-constrained date for the tsunami event. Māori pūrākau often provide even more detailed information (McFadgen and Goff, 2007; Goff and Chagué-Goff, 2015). Stories of taniwha may indicate big wave events that wreaked havoc on coastal communities, causing changes in settlement and local geomorphology (King and Goff, 2010; Goff and Chagué-Goff, 2015). Currently, cultural information is included for

14% of recorded tsunami events in the database, most of which have come from pūrākau ("New Zealand Palaeotsunami Database," 2017). The cultural information, alongside geomorphic and sedimentary information, provide key data for the generation of a robust and comprehensive palaeotsunami database for Aotearoa New Zealand (Goff et al., 2010).



## 6.2 International application of Aotearoa-NZ bicultural research frameworks

Indigenous communities around the world share many fundamental principles, including their interconnectedness with and inseparability from nature (Salmón, 2000; Wambrauw and Morgan, 2016). Other cultural values, such as environmental stewardship and sustainability are also common Indigenous values that guide ways of living and ways of knowing. Common values among Indigenous cultures enable and encourage transferability of established frameworks outside of the place where they have been developed. The three theoretical frameworks discussed in this review—He Poutama Whakamana, IBRLA and He Awa Whiria—can potentially be applied outside of Aotearoa-NZ, due to their flexible nature and adaptability for different

research groups and purposes. Likewise, the value-based models—the Mauri Model, the CFPS and SAM—can be modified to incorporate Indigenous values and priorities outside of the Aotearoa-NZ context, because the models are created with Indigenous groups on a case-by-case basis. Indigenous groups anywhere can specify which values they consider essential for the frameworks and models.

The Mauri Model, developed in Aotearoa-NZ, has been successfully applied in Papua, Indonesia to evaluate the

potential effects of a new agricultural development scheme in the Merauke regency, in the lowlands of Papua (Wambrauw and Morgan, 2014, 2016). Due to its ability to incorporate Indigenous and Western values, the Mauri Model was deemed an appropriate tool to assess the potential environmental and cultural impacts of the development scheme. The first step to successfully applying the model was to understand the new context in which it would be used. After confirming the Mauri Model would be appropriate, stakeholders for the project were selected, which included the Malind Anim Indigenous peoples.

The Mauri Model was adjusted to have a minimum value of -3 and a maximum value of +3 (rather than -2 and +2, respectively), based on local values and requirements. The results from using the Mauri Model indicated that the cultural values associated with the site would be denigrated if the development scheme proceeded. The Mauri Model provided semi-quantitative evidence that the development scheme would have serious negative impacts on the Malind Anim.

It is challenging to review the applicability of Aotearoa-NZ frameworks and models to international geomorphic

research because, to our knowledge, there are extremely few studies that explicitly use the frameworks to conduct geomorphic research outside of Aotearoa-NZ. However, we believe that there is great potential for these frameworks and models to be adapted outside of Aotearoa-NZ. The case of using the Mauri Model in Papua indicates that this model is transferrable, which suggests that the others could be as well. If the models are adapted appropriately and in accordance with local Indigenous communities' values and desires, we see no encumbrance to using these models in international geomorphic research.

## 6.3 The modern role of Earth surface science in society

There is a growing understanding that long-term sustainability on Earth is not achievable with monodisciplinary or reductionist scientific approaches (Pingram et al., 2019). Increasingly, geomorphology and Earth surface science are playing stronger roles in modern society and policy, guiding legislative action towards a more sustainable future. Sustainability is also at the core of many Indigenous cultures, which has enabled Indigenous knowledge and ways of life to persist for generations. We propose





that both Indigenous concepts and values and Westernised understandings of landscape processes have the potential to generate
significant changes in the way humans (including scientists) interact with the world. More so, if these two streams of
knowledge work together from the onset of a research project, there is the possibility of making discoveries that could not be
made by either approach alone.

As previously discussed, landmark policy achievements that consider both scientific and Indigenous concepts
emphasise the human and non-human elements of landscapes (Brierley et al., 2018; Aho, 2019; Pingram et al., 2019). These
policies prioritise sustainability by acknowledging the integrity of both geomorphic science and Indigenous knowledge. These
policies include legal personhood for rivers and the legal rights of nature (Brierley et al., 2018; O'Donnell and Talbot-Jones,
2018; Eckstein et al., 2019). Policies such as these provide opportunities for geomorphologists and Indigenous communities
to act as advocates for the landscape, which is a relatively novel approach to sustainable landscape management and interaction
within Westernised societies.

## 7. Conclusions and recommendations to geomorphologists

Incorporating Indigenous knowledge with Western science has the potential to bring mutual benefits to scientists, Indigenous
communities and governments. This review highlighted theoretical frameworks for including mātauranga Māori and Māori
value-based models into geomorphic research in Aotearoa-NZ. Each of the theoretical frameworks—He Poutama Whakamana,
IBRLA and He Awa Whiria—and value-based models—the Mauri Model, the Cultural Flow Preference Study and the
Sustainability Assessment Model—provide different benefits to scientists and Māori, and the most appropriate framework
selection for projects will occur on a case-by-case basis with Māori involvement. Though this review mostly focused on the
Aotearoa-NZ context, these frameworks are all capable of being applied in bicultural research contexts across the globe, so
long as they accurately reflect the values and knowledge of the local Indigenous peoples. We encourage geomorphologists
interested in working with Indigenous communities to consult with Indigenous peoples engagement support teams or
Indigenous studies departments at their local research institutes. Additionally, in Aotearoa-NZ, we encourage researchers
embarking on geomorphic research to consult iwi management plans and national funding guidelines for assistance in
identifying potential research avenues that may include mātauranga. The potential for including these frameworks in
geomorphic research is promising, particularly where such work overlaps with iwi aspirations.

We hope this review encourages and inspires geomorphologists to explore landscapes in Aotearoa-NZ and the world
through a bicultural lens, one that includes both Indigenous knowledge and modern scientific techniques to acknowledge and
respect the uniqueness of the world's landscapes. Using the approaches reviewed here have high potential to yield better
outcomes, as drawing from both knowledge systems will realise new understandings and solutions that neither body of
knowledge could reach in isolation.



**8. Author contribution statement**

Wilkinson and Hikuroa developed the original ideas and organisation of this review. Macfarlane provided translations for Māori terminology and developed ideas for theoretical framework analysis. Hughes contributed concepts that also improved framework and model analysis. Wilkinson prepared the manuscript with contributions from all co-authors.

The authors declare that they have no conflict of interest.

**9. Acknowledgements**

Financial support for publishing this manuscript was provided by the University of Canterbury School of Earth and Environment. The authors also acknowledge funding from QuakeCORE that has supported Wilkinson to pursue this topic as part of her PhD candidature. The authors would like to thank Dr. Timothy Stahl for his comments and feedback that greatly
improved previous versions of this manuscript. We also thank our reviewers for their constructive feedback that further strengthened this review.

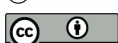



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





Wilkinson, C., and Macfarlane, A.: Braiding the Rivers of Geomorphology and Mātauranga Māori: A Case Study of
Landscape Healing in Koukourarata, Landscape Foundation, in press.

1010