# Peer review of "Mātauranga Māori in geomorphology: existing frameworks, case studies and recommendations for incorporating Indigenous knowledge in Earth science"

_Earth Surface Dynamics, 2020_

## Short Comment (SC1) · 4 Mar 2020

This paper provides a comprehensively researched and carefully drawn road map for conducting bicultural geoscience. After reviewing studies of indigenous knowledge and distinguishing Western science and matauranga as systems of knowledge production, the authors compare three theoretical frameworks for bicultural knowledge production, and then compare three more specific 'models' for weaving Maori knowledge and fluvial science together. These examples are extremely helpful, though the He Awa Whiria framework and the Sustainability Assessment Model are not detailed enough to enable a practical comparison against the others. The paper also describes New Zealand-

based geoscientists' specific obligations to support Maori knowledge through Vision Matauranga, and how that operates through funding structures. In assessing the different theoretical and practical approaches to weaving knowledges, the authors draw out implications for geomorphology both locally and internationally.

A unique strength of the paper is its careful treatment of matauranga Maori. Although I cannot evaluate the cultural validity of the authors' treatment of matauranga, I would say that it is among the more sophisticated and ethically nuanced social scientific treatments of matauranga that I have read. The authors clarify that the matauranga described in the paper (e.g. the concepts and values in the glossary) is not their own and that it belongs to the communities that have produced it. They urge that matauranga should not be conceived as simply a body of knowledge but as an entire system through which the world is known and experienced. The concept of whakapapa (translated as genealogy) is "a structured methodology for creating matauranga" (L179) and as such the question of who produces knowledge matters as much as what kind of knowledge is produced. The authors also usefully reinforce the point that "Scientists cannot rebuild or revitalise matauranga; that is for Maori to do (Broughton et al. 2015)". This nuance helps geoscientists to understand their roles as supporting Maori as matauranga producers. In the account of Te Ao Maori (the Maori world view) presented here, the social relations of knowledge shape its validity as much as the method by which it is produced. Together, these nuanced points about matauranga enable geoscientists to build respectful relationships (and respectful boundaries) with Maori communities.

The paper, while necessarily selective, is one of the most comprehensive of its kind while remaining concise, accessible, and focussed on the 'so what' for geoscience-trained researchers. The paper provides a useful guide for geoscientists (or even social scientists!) who are interested in conducting research in ways that seek to respect and enhance the dignity of indigenous peoples and their knowledges. The paper is effectively organised, well written, and tailored to its geoscience audience.

One big question that the paper made me reflect upon is this: can we do more to

convince geoscientists that indigenous knowledge has value beyond just 'filling in the historical record'? Within this paper and others – even those about matauranga specifically – the typical argument from scientists is that indigenous knowledge of ecological change can help to fill in historical data points and enable science to proceed as usual. This means that science per se does not need to change, and that indigenous knowledge can help them to do even better science as they have always known it. Many scientists can understand and agree with this argument, and Sections 1 and 2 of the paper convey this argument effectively. Yet this argument has the effect of placing indigenous knowledge under the evaluative lens of science, and creates an extractive and assimilative rather than respectful relation between science and indigenous knowledge. In this formulation matauranga must prove itself as a version of science to be considered valuable, whereas the reverse is not true. Is it possible to convince geoscientists to value and respect indigenous knowledge without making this 'instrumentalist conceit'?

---

## Referee Comment (RC1) · Carolina Londono (Referee) · 27 Mar 2020

General comments:

The paper presents a review of existing frameworks and models that have been used to incorporate Aotearoa Maori knowledge in New Zealand. It highlights case studies to exemplify how the frameworks work. It considers how the existing frameworks and studies apply to geomorphology and discuss the implications for studies outside of NZ. This is a high-quality review, it is well written and relevant. The frameworks presented should be a model for the US and the world where non-indigenous geoscientists wish to engage in research with indigenous peoples or their lands.

[Figure]

Specific comments I appreciate the words in the Maori language. But I found it taxing and distracting to go back and forth looking for the meaning. The authors should consider increasing the readability of the paper by including the English translation in parentheses, in some of the cases. Also, including a line or two justifying why using the words in Maori. I agree that they should be included, but the relevance of this choice may not be clear to everyone. Along with the terms, a phonetic guide would be useful.

A question I have is: What methods did the authors use for this paper? Did the authors conduct archival research? If so, please state it. On line 91, the authors mentioned permissions granted by the University to do the research. What did you have to ask permission for?

Technical corrections

Line 46, the authors talk about "a resurgence of sincere, respectful and reciprocal re-engagement between scientific and Indigenous communities". The word 'resurgence' implies that in the past, western and indigenous researchers collaborated in a respectful and reciprocal relationship. However, the history of colonialism shows that such a relationship has never existed. Instead, the western sciences have been extractive, non-reciprocal and disrespectful. The authors acknowledge this in lines 273-274. Thus, replace the words resurgence and re-engagement.

Line 57: Define "right of nature" to readers unfamiliar.

Move Table 1 so it appears after the first mention (it appears before so there's no context for it). Consider adding a guide for pronunciation (phonetic guide).

Line 176: What does it mean that: Whakapapa (...) fosters credibility by establishing connections between researchers and subjects?

Section 3.2. Consider making it shorter and clearly showing how this treaty connects to the frameworks.

Line 246: Just a comment, giving a river the legal personhood status is the way to go.

I celebrate; this!

Line 382-383: Could Fig. 3 be referenced there?

Line 386: What is Maori phenomena?

Line 426: How can conclusions be supported by both streams when one of the streams may lack the tools or paradigms of the other? What does it mean that both streams have to support findings? This is not clear to me.

Line 429: States that the method allows western science to stay true to the scientific method. Is this different from the other two? What do you mean when you say that there is no "hindrance" in using the scientific method for the other two?

Figure 4: This figure needs more explanation. For example, Whatwhat do the turquoise lines represent? Are we trying to connect the baskets? Do the arrows end in a particular place for a particular reason? And what do the horizontal double head arrows represent? And why using weaved baskets to represent both knowledges (i.e., western and Maori)

Paragraph starting in 549 states that non-Maori researchers could include Maori values. This raises questions for me. This could lead to cultural misrepresentation or cultural appropriation of knowledge. How are westerners going to interpret the Maori values when they are not part of that culture? I suggest revising this idea, and changing the wording to make it a REQUIREMENT of having a Maori researcher on the project, instead of a desirable situation.

615: Talks about "flexible" research methods. I'm concerned that this could translate as making science less rigorous or lowering its quality. I know that's not what is meant. I'd suggest changing 'flexible' to inclusive, adaptive or culturally responsive research methods.

Line 658: Besides adapting, or extrapolating, the Maori models to other parts of the globe, this article shows how researchers and indigenous peoples can develop frameworks and models particular for their culture. I would add that as a contribution.

---

## Referee Comment (RC2) · Anonymous Referee #2 · 30 Apr 2020

General Comments

This paper is a very useful review and analysis of bicultural research, with recommendations on how to better incorporate Indigenous knowledge (Matauranga Maori) in the science of geomorphology or in describing geomorphic processes. The paper should be accepted for publication, but with careful consideration of the following issues or suggestions for a moderate revision.

The title of the paper refers to "Matauranga Maori in geomorphology" or in other words the Knowledge held by Maori in the science of geomorphology or geomorphic processes. The second part of the title is confusing, and could be reworded to "Matau-

ranga Maori in Geomorphology: existing frameworks, case studies and recommendations for incorporating Indigenous Knowledge in earth science". The other interpretation of the first part of the title, which first drew me in, was the thought that the paper would review actual Matauranga Maori knowledge of geomorphic processes and phenomenon as local people. This knowledge is likely significant, as current occupiers and managers of landscapes, beyond just oral stories of past events or creation stories. The introduction of the paper could better differentiate these two versions of Matauranga Maori in geomorphology, and emphasise that the goal of the paper is to review the frameworks for knowledge incorporation in western science, rather than review the Indigenous geomorphic knowledge itself (but the brief review up front is helpful and insightful).

This in an important article needed to better inform geomorphologists of how to incorporate Indigenous knowledge in their research, or conversely, how to incorporate the science of geomorphology in education and the practical management of land by Indigenous people like the Maori . The latter could also be emphasised to an equal degree for balanced bicultural research, with suggestions on bidirectional education in contemporary Indigenous cultures that adapt to change.

All too often geomorphologists (and other scientists) ignore engagement with Indigenous communities and their traditional ownership of historic estates. They disrespect Indigenous rights to know of, control or guide, and/or participate in research on their traditional land, irrespective of current ownership or tenure or laws requiring it. This is a science version of continued colonisation and suppression. It should be emphasised to the reader that no matter if or how scientists involve Indigenous Knowledge in their proper research, they have an obligation at a minimum to engage with Indigenous people and custodians while conducting research on their traditional land, and most specifically ask permission to conduct the research on traditional land according to local protocols.

At a minimum this will help prevent geomorphologists from damaging cultural sites

during research, disrupting cultural norms or taboos, and widening the divide between scientific elites and local people. Asking research permission on traditional land is the first prerequisite, with adding Indigenous community members (or guides) to the team secondary, and gaining the use of Indigenous knowledge then tertiary.

The issue of Intellectual Property of Indigenous Knowledge also needs to be reviewed more in the paper. Often Indigenous knowledge is owned by the collective of multiple-generations (community), past, present and future. Having one or several Indigenous community members or leaders on a research group or board (paid or unpaid) does not automatically give permission to use or include collective Indigenous knowledge for scientific purposes, even if held in the mind and agreed to be shared by one person. Agreement from the collective is often needed, through a Memorandum of Understanding or Intellectual Property agreement with a Council of Elders, Tribal Council, or Indigenous Corporation, or others. This can become a sticky issue, and partially why some scientists often ignore the development of IP agreements. Regardless, this should become an official part of business by researchers around the world as required by funding agreements (Human Ethics even if not studying humans!), and national, regional, local and Indigenous governments. It would be great if the authors could convey some of these issues to readers, many of which are naïve to the issues.

Specific Comments

Overall, the paper is fairly long, with many sub-headings, and is easy to get lost within. It is written more like a PhD thesis chapter and review, and not a concise paper for journal publication and easy education. Please condense and remove any extraneous word, sentences, sections, or references, where possible? Avoid repeating references or ideas unless critical. Every word counts or distracts the reader. So sentence intros like "As discussed earlier" or "As previously mentioned" do not help, as one of many examples. As another of many examples Line 390 should be reduced "He Poutama Whakamana follows a kaupapa Maori research approach,. Kaupapa Maori , described in depth by Smith (2012), can be understood as research that is "culturally safe" and

**ESurfD**
that takes place within a Maori worldview (Irwin, 1994 as cited in Smith, 2012). Keep the sentences simple and straight forward and non-redundant.

Making it easy to read for a wide range of international geomorphologist will be key to having the information use and cited. The Table of Maori terms and names is very useful. However for the non-New Zealand reader, it is very hard to read the text and Maori terms and constantly go back to the table. It would be helpful to conduct two things: 1) make all Maori terms italics or otherwise to highlight to the reader the difference between English and written Maori (similar to what has been done with PNG language in the paper), and 2) at the end of key Maori words to have the short definition in brackets, like Iwi (tribe). This could be done at the location of first usage (which has been done in some places). Repeating it again at several key locations in the paper would also be helpful where important terms are used again. The authors do this for Matauranga Maori (Maori Indigenous knowledge), but not others like tangata whenua. The authors in places due this with commas, but the sentences get too complex. . . .. Line 237, For mana whenua, spiritual values of the Te Manahuna, the Mackenzie basin, are held as a priority to be conserved, which may be challenging to communicate to their partners. It would be easier to read as follows. For Mana whenua (people with with authority), spiritual values of the Te Manahuna (the Mackenzie basin) are held as a priority to be conserved, which may be challenging to communicate to their partners. Maori terms could also be capitalised, Iwi (tribe) to make stand out, if appropriate for written Maori ? Regardless, please make changes as if you were reading the paper as a German geomorphologist with basic English reading skills who wants to engage with African Indigenous Knowledge in her research.

Overall I had to read the paper twice to understand where it was all going and the big picture. The section titles and outline are key to improve upon. The sections headings are as follows with suggested additions and changes in italics to the titles below. Some headings could be deleted or combined.

1 Introduction

**ESurfD**
[Figure]

Please better define the difference between a Framework and Model earlier on in the paper. Overall these uses are very confusing to a new reader. The authors cover the difference better in section 5.2, but this needs to happen earlier in the paper (introduction) in a more concise and clear fashion. The authors mention 3 frameworks and models each, but there are lots of similarities and differences. In Table 2, a Framework is defined as a methodology, and Model is defined as a method. Theoretical vs actionable is key, but the Theoretical frameworks are actionable depending on the user and interpretation. Methodology as a general research strategy, and method as a tool to answer a question. In some place this use is even mixed up, such as Line 354 "The models proposed by Smith (1992, 2012) can be thought of as methodologies, or guiding principles. . ...". In this case and usage the sentence should read "The framework proposed by Smith (1992, 2012) can be thought of as methodologies, or guiding principles. . ...".Please educate the reader why they are labelled or grouped as is, both in the abstract, introduction, and also the main sections such as section 4 in paragraph Line 355 and 370, and in section 4.2. Section 5.2 does a better job at describing these differences.

In some locations the authors intermix geologic, geomorphic(ology) and earth science. Even in the title. And at times river science and health and ecology. The paper and journal focus is on geomorphology, perhaps leave it as that and omit the others. Geomorphology is pretty broad and inclusive. Just refer to the broader earth science when talking about wider applications, and the more specific sciences like river health and environmental flow where appropriate for the example reference.

Line 80. This sentence needs to be broken into two. We then introduce Te Ao Maori (the Maori world), discuss obligations of the New Zealand government to Maori , and present frameworks for conducting mixed-methods scientific research with iwi and hapu (tribes and family groupings—the principle political units with whom scientists engage) in Aotearoa-NZ in this space.

Line 84. This sentence is vague. We then provide case studies of framework development and recommendations for framework implementation in geomorphology research.
Line 287 paragraph is connected to the discussion in Line 300 paragraph in the next section. Repetitive and confusing to repeat. Please clarify and simplify or consolidate.

Figure 3. Make sure that this image is high enough resolution in print to be readable in a condensed format in a journal paper. Even in this full page format it is hard to read, and the journal may not print it as a full page.

Line 431 Knowledge of

Sections 5.1 Knowledge versus values (Revisited) and 5.2 Framework and Model recommendations for Geomorphology subdisciplines should be renamed, as the first really covers model application to capture values, while the second focuses on frameworks. Same with the Section 5 title, which focuses on both frameworks and models. It just gets confusing about what each paragraph or sub-section is referring to.

6.1 Direct benefits to geomorphology. Rather than just focusing on knowledge of physical events to benefit geomorphologist, the more common international benefit of working with Indigenous people is learning from their current intricate knowledge of the environment and physical and cultural and biological landscapes. If one wants to learn about all the springs in a catchment, who better to ask than local Indigenous people? Or locations of rock outcrops with valuable resources or tools? Or unique species isolated above geologic barriers? The paper missed out on a wealth of knowledge beyond past events.

Line 685 The key recommendation should be to encourage geomorphologists interested in working with Indigenous communities to consult directly with Indigenous communities and their self-governance institutions. There is a surprising level of diversity in governance capacity of Indigenous communities around the globe. Direct consultation is best, with support of other programs and experts of course where needed.

Overall, thank you for putting forward many ideas and recommendations to help scientists become better at engaging with Indigenous People.

**ESurfD**

Interactive
comment

---

## Author Response (AR2)

Clare Wilkinson
School of Earth and Environment
University of Canterbury
Christchurch, NZ

Attn: Professor Heather Viles
Associate Editor
Earth Surface Dynamics
23 May 2020

Dear Professor Viles,

Thank you for managing our manuscript "Mātauranga Māori in Geomorphology: existing frameworks, case studies and recommendations for Earth scientists" (manuscript number esurf-2020-5). We are grateful for the valuable reviews that we received from Carolina Londono and an anonymous referee. We hope the changes to our manuscript will be satisfactory for publication in Earth Surface Dynamics.

We thank the two referees for their constructive and insightful reviews. Based on their comments, as well as the short comment we received from Dr. Marc Tadaki, we felt the most important actions to take were to:

1. Increase readability of the text in terms of Māori language terms and English translations;
2. Strengthen the explanation of Figure 4 and the associated He Awa Whiria framework;
3. Better outline the goal of reviewing frameworks and models for weaving Indigenous knowledge with Western science in the Introduction;
4. Streamline the article by condensing and removing extraneous language.

On the following pages, we address general comments from reviewers and provide a table that includes specific and technical reviewer comments, our explanation for changing or not changing the original text, and any modifications made. We also provide two .pdf versions of our updated manuscript: one with tracked changes and one without.

We also became aware of additional relevant literature since the original submission data of our manuscript and felt it appropriate to add in these references:

1. Cano Pecharroman, L.: Rights of Nature: Rivers That Can Stand in Court, Resources, 7, 13 pp., doi:10.3390/resources7010013, 2018.
2. Kauffman, C.M. and Martin, P.L.: Constructing Rights of Nature Norms in the US, Ecuador, and New Zealand, Global Environmental Politics, 18, 43-62, doi:10.1162/glep_a_00481, 2018.
3. Maxwell, K.H., Ratana, K., Davies, K.K., Taiapa, C., and Awatere, S.: Navigating towards marine co-management with Indigenous communities on-board the Waka-Taurua, Marine Policy, 111, 4 pp., doi:10.1016/j.marpol.2019.103722, 2020.
4. Wilcock, D. and Brierley, G.: It's about time: extending time-space discussion in geography through use of 'ethnogeomorphology' as an education and communication tool, Journal of Sustainability Education, 3, 2012.
5. Wilcock, D., Brierley, G., and Howitt, R.: Ethnogeomorphology: Progress in Physical Geography, doi:10.1177/0309133313483164, 2013.

Again, thank you for managing our manuscript and for facilitating the involvement of our reviewers; we greatly appreciate their feedback.

Sincerely,

Clare Wilkinson, on behalf of the authorship team.

Clare Wilkinson
School of Earth and Environment
University of Canterbury
Christchurch, NZ

**Responses to Reviewer 1, Carolina Londono**

General Comments

*The paper presents a review of existing frameworks and models that have been used to incorporate Aotearoa Maori knowledge in New Zealand. It highlights case studies to exemplify how the frameworks work. It considers how the existing frameworks and studies apply to geomorphology and discuss the implications for studies outside of NZ. This is a high-quality review, it is well written and relevant. The frameworks presented should be a model for the US and the world where non-indigenous geoscientists wish to engage in research with indigenous peoples or their lands.*
Thank you.

Specific and technical comments

| Reviewer Comment | Original Line Number | Author Comment | Author Revision | New Line Number |
|---|---|---|---|---|
| I appreciate the words in the Maori language. But I found it taxing and distracting to go back and forth looking for the meaning. | Throughout | Agree | We have added short English translations for Māori terms where appropriate. | Throughout |
| Also, including a line or two justifying why using the words in Maori. | N/A | Respectfully Disagree— no change required | We use words in *te reo* Māori (Māori language) to be inclusive throughout our review. We intend to demonstrate—rather than justify—our dedication to weaving Māori knowledge with Western approaches, and one way to honour Māori is to learn and promote their language. | N/A |
| What methods did the authors use for this paper? | N/A | Sentence added | Added: "We used archival research, review and *wānanga* (discussion) to conduct this research." | 85 |
| On line 91, the authors mentioned permissions granted by the University to do the research. What did you have to ask permission for? | 91 | Sentence added | At the University of Canterbury (PI Wilkinson's institute), all research conducted by staff or students that involves Māori groups in any way must be approved by the University's Human Ethics Committee. This literature review is part of Wilkinson's PhD research, which includes interviews and face-to-face interactions with | 91-95 |

| | | | individuals from different Māori iwi (tribes). Therefore, we had to gain ethics approval before conducting this research. We also felt it is important for readers to know that we complied by the policy of asking permission to discuss mātauranga, and the information provided in the review is not something we can claim as our own.

Moreover, it is important to note that gaining permission through Human Ethics Committees helps to safeguard the Intellectual Property of Indigenous peoples. This point was raised by Reviewer 2, and we added a sentence to indicate that our Ethics approval acknowledges our obligation as researchers to respect and protect that Intellectual Property. | |
|---|---|---|---|---|
| Thus, replace the words resurgence and re-engagement. | 46 | Change made | Changed "resurgence" to "emergence" and "re-engagement" to "engagement"

Note: Also in the abstract (line 10) we replaced "experiencing a resurgence" with "emerging" | 45-46 |
| Define "right of nature" to readers unfamiliar. | 57 | Change made | Definition added | 56 |
| Move Table 1 so it appears after the first mention (it appears before so there's no context for it). Consider adding a guide for pronunciation (phonetic guide) | | Agree | Table 1 now has a phonetic guide and appears below the first mention | Page 6, near line 160 |
| What does it mean that: Whakapapa (…) fosters credibility by | 176 | Change made | We changed "subjects" to "research objectives". By subjects, we meant the subject | 170 |

| | | | of the researcher's research. We hope "research objectives" clarifies any uncertainty here. | |
|---|---|---|---|---|
| establishing connections between researchers and subjects? | | | | |
| Section 3.2. Consider making it shorter and clearly showing how this treaty connects to the frameworks. | | Partially agree | We felt this section was important for highlighting modern interpretations of the Treaty of Waitangi and how it is being used in research and engagement today. This section is intended to illustrate that the principles of the treaty are being used to guide transformative policy and management schemes, so why can't we also use the principles in geomorphic research?

We also felt it was important to establish the context for research in Aotearoa-NZ, which is guided by this governing document.

Having said that, we condensed section 3.2 by removing section heading 3.2.1 and changing the 3.2 section heading to reflect what was in 3.2.1.

We added a sentence in the IBRLA framework section (line 410) highlighting the Treaty of Waitangi principles woven throughout the framework. | Section 3.2 (beginning line 199) |
| Just a comment, giving a river the legal personhood status is the way to go. I celebrate; this! | 246 | Agree | Thank you! | |
| Could Fig. 3 be referenced there? | 382-383 | Agree | Done | 374 |
| What is Maori phenomena? | 386 | Agree | individuals, culturally significant landscapes, values—we have added this in | 376-377 |

Clare Wilkinson
School of Earth and Environment
University of Canterbury
Christchurch, NZ

| How can conclusions be supported by both streams when one of the streams may lack the tools or paradigms of the other? What does it mean that both streams have to support findings? This is not clear to me. | 426 | | We have changed the language we use here to be more consistent with the language used in our review of the other two frameworks. We now say: "Ultimately, when research conclusions are drawn, they must represent co-creation of knowledge using both streams." | 423 |
|---|---|---|---|---|
| States that the method allows western science to stay true to the scientific method. Is this different from the other two? What do you mean when you say that there is no "hindrance" in using the scientific method for the other two? | 429 | Changes made | In regard to the He Poutama Whakamana and IBRLA frameworks, we changed the use of "hindrance" to "maintaining integrity". All frameworks allow the scientific method to be used.

Historically, one of the biggest reasons for scientists to hesitate to include Indigenous knowledge in their research was the concern that Indigenous knowledge might interfere with the scientific method. We feel it is important to demonstrate that Indigenous knowledge and Western science can work together without undermining each other; we wanted to be explicit about the ability to still use the scientific method while weaving Indigenous knowledge into research projects.

We have removed the explicit mention of the scientific method in the He Awa Whiria framework and instead use terms such as "the Western science paradigm" and "Western science analysis". We maintain our usage of the scientific method in the He | Paragraph beginning line 424 |

| | | | Poutama Whakamana and IBRLA frameworks. | |
|---|---|---|---|---|
| Figure 4: This figure needs more explanation. For example, What do the turquoise lines represent? Are we trying to connect the baskets? Do the arrows end in a particular place for a particular reason? And what do the horizontal double head arrows represent? And why using weaved baskets to represent both knowledges (i.e., western and Maori) | Figure 4 | Agree | Thank you for this comment. We have added a better explanation of the imagery in the figure and why those symbols are significant (including the baskets). In the caption to Figure 4, we now state that the turquoise lines represent knowledge exchange and development throughout the research programme. | Section 4.1.3 and Figure 4 (beginning line 411) |
| Paragraph starting in 549 states that non-Maori researchers could include Maori values. This raises questions for me. This could lead to cultural misrepresentation or cultural appropriation of knowledge. How are westerners going to interpret the Maori values when they are not part of that culture? I suggest revising this idea, and changing the wording to make it a REQUIREMENT of having a Maori researcher on the project, instead of a desirable situation. | 549- | Partially Agree | Thank you for this valuable comment. We believe that "requiring" Māori participation in research runs the risk of perpetuating colonizing practices. We believe it is best for Māori communities to choose their level of involvement. The text relating to this comment remains unchanged. We have however added an indication of this important point in a later part of the text, where we discuss resources for initiating research projects with Māori (see lines 593-597). | 572; 593-597 |
| Talks about "flexible" research methods. I'm concerned that this could translate as | 615 | Agree | Thank you for this comment—changed "flexible" to "adaptive" as you suggest. | 620 |

| | | | | |
|---|---|---|---|---|
| making science less rigorous or lowering its quality. I know that's not what is meant. I'd suggest changing 'flexible' to inclusive, adaptive or culturally responsive research methods. | | | | |
| Besides adapting, or extrapolating, the Maori models to other parts of the globe, this article shows how researchers and indigenous peoples can develop frameworks and models particular for their culture. I would add that as a contribution. | 658 | Agree | Thank you—we have added this in. | 659-661 |

**Responses to Reviewer 2, anonymous**

General Comments

*The title of the paper refers to "Matauranga Maori in geomorphology" or in other words the Knowledge held by Maori in the science of geomorphology or geomorphic processes. The second part of the title is confusing, and could be reworded to "Matauranga Maori in Geomorphology: existing frameworks, case studies and recommendations for incorporating Indigenous Knowledge in earth science". The other interpretation of the first part of the title, which first drew me in, was the thought that the paper would review actual Matauranga Maori knowledge of geomorphic processes and phenomenon as local people. This knowledge is likely significant, as current occupiers and managers of landscapes, beyond just oral stories of past events or creation stories. The introduction of the paper could better differentiate these two versions of Matauranga Maori in geomorphology, and emphasise that the goal of the paper is to review the frameworks for knowledge incorporation in western science, rather than review the Indigenous geomorphic knowledge itself (but the brief review up front is helpful and insightful).*

Thank you—we certainly don't want the title to be misleading in anyway. We have taken up your suggestion to call the review "Mātauranga Māori in geomorphology: existing frameworks, case studies and recommendations for incorporating Indigenous knowledge in Earth science".

*This in an important article needed to better inform geomorphologists of how to incorporate Indigenous knowledge in their research, or conversely, how to incorporate the science of geomorphology in education and the practical management of land by Indigenous people like the Maori . The latter could*

Clare Wilkinson
School of Earth and Environment
University of Canterbury
Christchurch, NZ

*also be emphasised to an equal degree for balanced bicultural research, with suggestions on bidirectional education in contemporary Indigenous cultures that adapt to change.*

We agree that this is an important issue, but we do not feel that discussing it is appropriate for what we are trying to accomplish with our review. The frameworks and models we discuss do seek to achieve balanced bicultural geomorphology research, but we feel that delving into bidirectional *education* is beyond the scope of this review.

*All too often geomorphologists (and other scientists) ignore engagement with Indigenous communities and their traditional ownership of historic estates. They disrespect Indigenous rights to know of, control or guide, and/or participate in research on their traditional land, irrespective of current ownership or tenure or laws requiring it. This is a science version of continued colonisation and suppression. It should be emphasised to the reader that no matter if or how scientists involve Indigenous Knowledge in their proper research, they have an obligation at a minimum to engage with Indigenous people and custodians while conducting research on their traditional land, and most specifically ask permission to conduct the research on traditional land according to local protocols.*
*Asking research permission on traditional land is the first prerequisite, with adding Indigenous community members (or guides) to the team secondary, and gaining the use of Indigenous knowledge then tertiary.*

We agree that there should be a need for researchers to consider how their research may be applicable to/of interest to Indigenous communities. What we have done in our review is stress the need to engage with Indigenous groups when appropriate, and document how it is done in Aotearoa-NZ. We wish to provide general guidance to researchers that will encourage them to discover their own local engagement procedures, without being overly prescriptive. We feel it is most important for researchers to be guided by the experts in their local area. Therefore, we respectfully choose to maintain the way we have discussed engagement with Indigenous communities.

*The issue of Intellectual Property of Indigenous Knowledge also needs to be reviewed more in the paper. Often Indigenous knowledge is owned by the collective of multiple generations (community), past, present and future. Having one or several Indigenous community members or leaders on a research group or board (paid or unpaid) does not automatically give permission to use or include collective Indigenous knowledge for scientific purposes, even if held in the mind and agreed to be shared by one person. Agreement from the collective is often needed, through a Memorandum of Understanding or Intellectual Property agreement with a Council of Elders, Tribal Council, or Indigenous Corporation, or others. This can become a sticky issue, and partially why some scientists often ignore the development of IP agreements. Regardless, this should become an official part of business by researchers around the world as required by funding agreements (Human Ethics even if not studying humans!), and national, regional, local and Indigenous governments. It would be great if the authors could convey some of these issues to readers, many of which are naïve to the issues.*

Thank you for this comment. We feel that this point is perhaps a bit too far down the chain of engagement to include in our paper at length. We agree that this is incredibly important and have added a sentence in our introduction explaining why we had to gain Human Ethics permission to conduct our research, hoping that it illustrates this important step. Our paper aims to encourage geoscientists to embark on research journeys with Indigenous groups and, as we have stressed, we implore researchers to discuss their research ideas *early* with staff at their University or Research institute who are skilled in appropriate engagement processes. Conversations about the IP of Indigenous knowledge will stem from those discussions with cultural engagement advisors. However, we greatly value and appreciate this

Clare Wilkinson
School of Earth and Environment
University of Canterbury
Christchurch, NZ

comment. The last 3 sentences of our introduction now read: "We acknowledge that the *mātauranga* presented here is not our own, and that we have gained approval through the Human Ethics Committee at the University of Canterbury (Christchurch, NZ) to conduct this research. In all cases, including our own, this approval is required in order to respect the Intellectual Property of Indigenous peoples. We herein acknowledge the *mana whenua* (traditional authorities) of Aotearoa-NZ as the rightful holders of *mātauranga*." The 2nd of the 3 provided sentences is new and the 1st and 3rd are from the original manuscript.

*The section titles and outline are key to improve upon. The sections headings are as follows with suggested additions and changes in italics to the titles below. Some headings could be deleted or combined.*

Thank you for these suggestions (we have moved this comment from the specific/technical corrections to here, where it is easier to address). We have made some changes where we agree that your suggestion is appropriate. We have maintained the original form of some headings where we feel further text in the heading is clunky. We removed one section heading (3.2.1) but changed the 3.2 section heading to reflect what was previously in 3.2.1. Because this is a review, we do feel the need to maintain our heading and subheading structure, so that the content of each section is clear.

| Reviewer Suggestion | Author comment | Current form |
| --- | --- | --- |
| *1 Introduction* | No change required | 1 Introduction |
| *2 Overview of International research at the interface of Indigenous knowledge and science* | No change required | 2 Overview of international research at the interface of Indigenous knowledge and geoscience |
| *3 Mixed-method geoscience research in contemporary Aotearoa-NZ* | No change required | 3 Mixed-method geoscience research in contemporary Aotearoa-NZ |
| *3.1 Te Ao Maori (the Maori worldview)* | No change required | 3.1 *Te Ao Māori* (the Māori worldview) |
| *3.1.1 Whakapapa and tikanga (Validity through ancestry)* | Changed | 3.1.1 *Whakapapa* and *tikanga* (validity through ancestry) |
| *3.1.2 Matauranga Maori (Indigenous Knowledge)* | Changed | 3.1.2 *Mātauranga* Māori (Māori knowledge) |
| *3.1.3 Kaitiakitanga (Well-being of people and environment)* | Changed | 3.1.3 *Kaitiakitanga* (Well-being of people and environment) |
| *3.2 Obligations of the Aotearoa New Zealand government to Maori* | No change required | 3.2 Obligations of the Aotearoa New Zealand government to Maori through the Treaty of Waitangi |
| *3.2.1 The Treaty of Waitangi (Maori and Crown as legal partners)* | Section header removed | -- |
| *3.2.2 The Treaty in practice* | Changed subheading number | 3.2.1 The Treaty in practice |
| *3.2.2.1 Te Manahuna Aoraki Project (Government Consolation)* | No change required (except subheading number) | 3.2.1.1 *Te Manahuna Aoraki* Project |

| | | |
|---|---|---|
| *3.2.2.2 Te Awa Tupua (Rivers at Legal People)* | No change required (except subheading number) | 3.2.1.2 *Te Awa Tupua* |
| *3.3 Woven spacesâ˘Tthe interface of Matauranga Maori and science* | No change required | 3.3 Woven spaces at the interface of *mātauranga* Māori and science |
| *3.3.1 The relationship between Matauranga and science* | No change required | 3.3.1 The relationship between *mātauranga* and science |
| *3.3.1.1 Indigenous knowledge versus values* | Changed | 3.3.1.1 Indigenous values |
| *3.3.2 Mutual research needs and benefits (Indigenous Management Plans)* | Slight change | 3.3.2 Identifying mutual research needs and benefits |
| *3.3.3 Potential challenges and risks of conducting research at the cultural interface* | Changed | 3.3.3 Potential challenges and risks of conducting research at the cultural interface |
| *4. Frameworks and models for incorporating Matauranga Maori alongside in geomorphic research* | No change required | 4. Frameworks and models for incorporating *mātauranga* Māori alongside in geomorphic research |
| *4.1 Theoretical Frameworks (Matauranga Maori in geomorphic research)* | No change required | 4.1 Theoretical frameworks for including *mātauranga* Māori in geomorphic research |
| *4.1.1 He Poutama Whakamana (Mirror-images of knowledge and understanding)* | Changed | 4.1.1 *He Poutama Whakamana* (mirror-images of knowledge and understanding) |
| *4.1.2 IBRLA (initiation, benefits, representation, legitimation, accountability)* | Changed | 4.1.2 IBRLA (initiation, benefits, representation, legitimation, accountability) |
| *4.1.3 He Awa Whiria (A Braided Rivers Approach)* | Changed | 4.1.3 *He Awa Whiria* (a braided rivers approach) |
| *4.2 Models (Step-By-Step Guide of Including Maori values in geomorphic research)* | No change required | 4.2 .2 Models for including Māori values in geomorphic research |
| *4.2.1 Mauri model (Sustainability and Cultural Bonds to the Environment)* | No change required | 4.2.1 *Mauri* model |
| *4.2.1.1 Transferability to geomorphology (Mauri model)* | Changed | 4.2.1.1 Transferability to geomorphology (*Mauri* model) |
| *4.2.2 Cultural Flow Preference Study (Cultural Practices and River Flow)* | No change required | 4.2.2 Cultural Flow Preference Study |
| *4.2.2.1 Transferability to geomorphology (Cultural Flow)* | Changed | 4.2.2.1 Transferability to geomorphology (CFPS) |
| *4.2.3 Sustainability Assessment Method (Values Associated with Waterway Health)* | No change required | 4.2.3 Sustainability Assessment Method |

Clare Wilkinson
School of Earth and Environment
University of Canterbury
Christchurch, NZ

| | | |
|---|---|---|
| *4.2.3.1 Transferability to geomorphology (Sustainability Assessment)* | Changed | 4.2.3.1 Transferability to geomorphology (SAM) |
| *5. Critical assessment of existing frameworks and models in different conditions* | No change required | 5. Critical assessment of existing frameworks and models in different conditions |
| *5.1 Knowledge versus values (Revisited)* | Changed | 5.1 Framework recommendations for subdisciplines |
| *5.2 Framework and Model recommendations for Geomorphology subdisciplines* | No change required | 5.2 Model application to include Indigenous values |
| *5.3 Guiding resources for initiating projects in Aotearoa-NZ* | No change required | 5.3 Guiding resources for initiating projects in Aotearoa-NZ |
| *6. Lessons for the international geomorphology community* | No change required | 6. Lessons for the international geomorphology community |
| *6.1 Direct benefits to geomorphology* | No change required | 6.1 Direct benefits to geomorphology |
| *6.2 International application of Aotearoa-NZ bicultural research frameworks* | Changed | 6.2 International application of Aotearoa-NZ bicultural research frameworks and models |
| *6.3 The benefit of Indigenous Knowledge and Geomorphology Science in Society* | Changed | 6.3 Benefits of Indigenous knowledge and geomorphology to society |
| *7. Conclusions and recommendations to geomorphologists* | Unchanged | 7. Conclusions and recommendations to geomorphologists |

Specific and technical comments

| Reviewer Comment | Original Line Number | Author Comment | Author Revision | New Line Number |
|---|---|---|---|---|
| Overall, the paper is fairly long, with many sub-headings, and is easy to get lost within…Please condense and remove any extraneous word, sentences, sections, or references, where possible? | N/A | Changes made | We have removed some repeat references and unnecessary words/sentences/phrases. | Throughout |

Clare Wilkinson
School of Earth and Environment
University of Canterbury
Christchurch, NZ

| So sentence intros like "As discussed earlier" or "As previously mentioned" do not help, as one of many examples. As another of many examples Line 390 should be reduced "He Poutama Whakamana follows a kaupapa Maori research approach,. Kaupapa Maori , described in depth by Smith (2012), can be understood as research that is "culturally safe" and that takes place within a Maori worldview (Irwin, 1994 as cited in Smith, 2012). Keep the sentences simple and straight forward and non-redundant. | N/A | Changes made | We removed as many sentence intros like this as we felt appropriate. | Throughout |
|---|---|---|---|---|
| The Table of Maori terms and names is very useful. However for the non-New Zealand reader, it is very hard to read the text and Maori terms and constantly go back to the table. It would be helpful to conduct two things: 1) make all Maori terms italics or otherwise to highlight to the reader the difference between English and written Maori (similar to what has been done with PNG language in the paper), and 2) at the end of key Maori words | Throughout | Agree | We have added short English translations for Māori terms where appropriate, and have italicised Māori terms. | Throughout |

Clare Wilkinson
School of Earth and Environment
University of Canterbury
Christchurch, NZ

| | | | | |
|---|---|---|---|---|
| to have the short definition in brackets, like Iwi (tribe). | | | | |
| The authors in places due this with commas, but the sentences get too complex. . ... Line 237, For mana whenua, spiritual values of the Te Manahuna, the Mackenzie basin, are held as a priority to be conserved, which may be challenging to communicate to their partners. It would be easier to read as follows. For Mana whenua (people with with authority), spiritual values of the Te Manahuna (the Mackenzie basin) are held as a priority to be conserved, which may be challenging to communicate to their partners. | Throughout | Agree | We have made the appropriate change.

Note: we also changed a similar occurrence of comma and em dash usage in the abstract to include just parentheses.

Note: again, we changed a similarly clunky sentence in original manuscript lines 99-102. | Throughout |
| Maori terms could also be capitalised, Iwi (tribe) to make stand out, if appropriate for written Maori ? | Throughout | Respectfully disagree | We have italicised all Māori terms to make them stand out. | Throughout |
| Please better define the difference between a Framework and Model earlier on in the paper. Overall these uses are very confusing to a new reader. The authors cover the difference better in section 5.2, but this needs to happen earlier in the paper (introduction) in a | Throughout | Agree | We provide a short definition of framework and model in the introduction (similar to the definitions included in Table 2).

We also provide more explicit definitions of "framework" and "model" at the beginning of section 4. | Section 4, beginning line 343 |

| | | | | |
|---|---|---|---|---|
| more concise and clear fashion. The authors mention 3 frameworks and models each, but there are lots of similarities and differences. In Table 2, a Framework is defined as a methodology, and Model is defined as a method. Theoretical vs actionable is key, but the Theoretical frameworks are actionable depending on the user and interpretation. Methodology as a general research strategy, and method as a tool to answer a question. | | | | |
| In some place this use [of framework and model] is even mixed up, such as Line 354 "The models proposed by Smith (1992, 2012) can be thought of as methodologies, or guiding principles. . ...". In this case and usage the sentence should read "The framework proposed by Smith (1992, 2012) can be thought of as methodologies, or guiding principles. . ...". | Throughout | Changes made | Thank you for this helpful comment. We have made sure that we do not mix up the usage of "framework" and "model" in the revised manuscript. | Throughout |
| Please educate the reader why they are labelled or grouped as is, both in the abstract, introduction, and also the main sections such as section 4 in | Throughout | Changes made | We have included more explicit definitions/explanations of the use of "framework" and "model". | Throughout |

| paragraph Line 355 and 370, and in section 4.2. Section 5.2 does a better job at describing these differences. | | | | |
|---|---|---|---|---|
| In some locations the authors intermix geologic, geomorphic(ology) and earth science. Even in the title. And at times river science and health and ecology. The paper and journal focus is on geomorphology, perhaps leave it as that and omit the others. Geomorphology is pretty broad and inclusive. Just refer to the broader earth science when talking about wider applications, and the more specific sciences like river health and environmental flow where appropriate for the example reference. | Throughout | Changes made | We have clarified our use of these terms and make sure we use the appropriate term in each location.

Note: We reorganised the paragraph beginning on Line 142 so that the mention of ecological studies is later in the paragraph rather than at the beginning. This has the effect of showing ecology is not the main topic of the paragraph, while still highlighting the importance of mentioning that Indigenous knowledge has been incorporated into ecology studies and that geomorphology might be imbedded in those studies. | Throughout |
| This sentence needs to be broken into two. We then introduce Te Ao Maori (the Maori world), discuss obligations of the New Zealand government to Maori , and present frameworks for conducting mixed-methods scientific research with iwi and hapu (tribes and family groupings—the principle political units with whom scientists | 80-83 | Agree | The sentences now read: We then introduce *Te Ao Māori* (the Māori world) and some Māori concepts relevant to geomorphology. We discuss obligations of the New Zealand government to Māori groups (i.e. *iwi* and *hapū*, tribes and sub-tribes, which are the principle political units with whom scientists engage in Aotearoa-NZ). We present three theoretical frameworks | 79-83 |

| | | | | |
|---|---|---|---|---|
| en- ˘ gage) in Aotearoa-NZ in this space. | | | (methodologies or general research strategies) and three value-based models (methods for answering research questions) for conducting mixed-method bicultural research. | |
| This sentence is vague. We then provide case studies of framework development and recommendations for framework implementation in geomorphology research. | 83-84 | Somewhat agree; changes made | We then provide case studies of model development and recommendations for implementation in geomorphology research. | 83-84 |
| Line 287 paragraph is connected to the discussion in Line 300 paragraph in the next section. Repetitive and confusing to repeat. Please clarify and simplify or consolidate. | 287-305 | Agree | We revised these two paragraphs so that the first is more focused on the relationship between mātauranga and science while the second is more focused on Indigenous knowledge and values. The second paragraph is now more concise. | 283-298 |
| Figure 3. Make sure that this image is high enough resolution in print to be readable in a condensed format in a journal paper. Even in this full page format it is hard to read, and the journal may not print it as a full page. | Figure 3, page 16 | | Thank you—it is 400 dpi (will discuss this further with the associate editor if necessary). | Figure 3, page 16 |
| Knowledge of | 431 | Changed | Changed to "allowing the two knowledge streams to operate…" | 428 |
| Sections 5.1 Knowledge versus values (Revisited) and 5.2 Framework and Model recommendations for Geomorphology subdisciplines should | 538-578 | Agree | We removed the original manuscript section with the heading " 5.1 Knowledge and values revisited" and distributed the information between the revised sections 5.1 | Section 5, starting line 529 |

| | | | | |
|---|---|---|---|---|
| be renamed, as the first really covers model application to capture values, while the second focuses on frameworks. Same with the Section 5 title, which focuses on both frameworks and models. It just gets confusing about what each paragraph or sub-section is referring to. | | | and 5.2. These revised sections are: "5.1 Framework recommendations for subdisciplines" and "5.2 Model application to include Indigenous values"

Note: We have also changed the heading for section 5 to: "5. Embarking on the bicultural research journey" to better reflect the sections that fall beneath it. | |
| 6.1 Direct benefits to geomorphology. Rather than just focusing on knowledge of physical events to benefit geomorphologist, the more common international benefit of working with Indigenous people is learning from their current intricate knowledge of the environment and physical and cultural and biological landscapes. If one wants to learn about all the springs in a catchment, who better to ask than local Indigenous people? Or locations of rock outcrops with valuable resources or tools? Or unique species isolated above geologic barriers? The paper missed out on a wealth of knowledge beyond past events. | 605- | Agree | Added in a few sentences to the second paragraph in this section to talk about contemporary Indigenous knowledge guiding geomorphic research. In the period of time between submitting our original manuscript and receiving reviews, we became aware of a publication by Wilcock et al. (2013) that discusses a concept they call 'ethnogeomorphology'. We briefly discuss this concept here to further address your comment. | Section 6.1, specifically lines 612-619. |

Clare Wilkinson
School of Earth and Environment
University of Canterbury
Christchurch, NZ

| The key recommendation should be to encourage geomorphologists interested in working with Indigenous communities to consult directly with Indigenous communities and their self-governance institutions. There is a surprising level of diversity in governance capacity of Indigenous communities around the globe. Direct consultation is best, with support of other programs and experts of course where needed. | 685 | Respectfully disagree | We believe that consultation with engagement support teams is the best way for geomorphologists to begin a bicultural research journey. The reason for this is because, as you state, there is a wide diversity in governance capacity of Indigenous communities around the globe, meaning that they will all have different expectations surrounding engagement protocols. We cannot provide specific engagement advice that would suit all Indigenous communities around the globe. Therefore, we advise researchers to talk to people at their own institutions who are knowledgeable about engagement protocols in their local area.

In many cases, Human Ethics must be approved before researchers can engage with Indigenous communities. Cultural advisors at universities and research institutes will be able to advise researchers on how to gain ethics approval. In our experience, there are many steps that must occur first before researchers directly engage with Indigenous groups. | N/A |
|---|---|---|---|---|

[revised manuscript text omitted]

theoretical frameworks (methodologies or general research strategies) and three value-based models (methods for answering
research questions) for conducting mixed-methodbicultural s scientific research with *iwi* and *hapū* (tribes and family
groupings the principle political units with whom scientists engage) in Aotearoa NZ in this space. We then then provide case
studies of framework model development and recommendations for framework implementation in geomorphology research.
Finally, we provide direct examples of including Indigenous knowledge in geomorphic research and discuss how the
frameworks and models reviewed here can be applied outside of the Aotearoa-NZ context. We used archival research, review
and *wānanga* (discussion) to conduct this research. We believe that the scientific world may learn some valuable lessons from
Aotearoa NZ about how Indigenous knowledge and geomorphology can work together to create new and innovative
understandings about how to live with and learn about Earth surface systems.

The authors assert that there is no expectation that *mātauranga* be given away by *iwi* (tribes) and *hapū* (sub-tribes) to scientists.  Scientists alone cannot

100 rebuild or revitalise *mātauranga*; that is for Māori to do (Broughton et al., 2015).  We uphold that the geoscience community is primed to contribute to further reinvigoration of *mātauranga* by welcoming it alongside science for greater understanding of Earth surface phenomena. Our intentions for this review are to encourage inclusion of Indigenous knowledge and values for guiding scientific research. We acknowledge that

105 the *mātauranga* presented here is not our own, and that we have gained approval through the Human Ethics Committee at the University of Canterbury (Christchurch, NZ) to conduct this research. In all cases, including our own, this approval is required in order to protect the Intellectual Property of Indigenous peoples. We herein acknowledge the *mana whenua* (traditional authorities) 
[revised manuscript text omitted]
., 2018). After Māori settling settled in Aotearoa-NZ many centuries ago (Hikuroa, 2017), Māori formed distinct groups emerged (today, about 40 *iwi* and hundreds of *hapū*) that , all of which built their identity from the surrounding mountains, lakes and rivers (Ruru, 2018). These tribal identities have implications for *mātauranga-a-iwi* (*iwi*-specific *mātauranga*), tribal ancestry, credibility and *iwi*-specific guardianship of tribal lands. Glossaries of Māori words (Table 1) and key English terminologies used in this paper (Table 2) are provided for reference.

Table 1: Glossary of Māori terms (as used in this paper)

| *TermArohatanga* | *Phonetic Guide* | *Care, respect, loveDescription* |
|---|---|---|
| Arohatanga | Ah-ror-ha-tah-nga | Care, respect, love |
| Atua | Ah-two-ah | Departmental gods, energies |
| Hapū | Hah-pooh | Sub-tribe |
| Hine-Titama | He-neh-Tea-tah-mah | The first human, a woman |
| Io-Matua-Kore | Eeyore-Mah-two-ah-Ko-reh | The supreme 'first' being in Māori cosmology |
| Iwi | E (as in letter e)-we | Tribe |
| Kaitiaki/kaitiakitanga | Kay (as in **kay**ak)-tea-ah-key/ Kay (as in **kay**ak)-tea-ah-key-tah-nga | Guardian and the act of guardianship; principle of intergenerational sustainability and the practices to achieve it |
| Kete | Keh-teh | Basket |
| Ki Uta Ki Tai | Key Oo (as in b**oo**t)-tah Key Tie | Literally Concept expressing the importance of catchments extending from the mountains to the se'To Mountain To Sea', this is a Maori holistic philosophy that considers the environment in its entirety, expressing the importance of catchments extending from the mountains to the sea |
| Mahinga kai | Mah-he-nga kay (as in **kay**ak) | Traditional food gathering practices and places |
| Mana | Mah-nah | Authority, prestige |

| | | |
|---|---|---|
| Mana whenua | Mah-nah Feh-nu-ah | People with traditional authority over the land |
| Manaakitanga | Mah-nah-ah-key-tah-nga | Acts of caring for and giving |
| Māramatanga | Mah-rah-mah-tah-nga | Enlightenment, understanding, a phase in which knowledge can be applied |
| Mātauranga Māori | Mah-tow-rah-nga Mah-or-ree | Knowledge, culture, values and worldview held by Māori, the Indigenous peoples of Aotearoa New Zealand |
| Mātauranga-a-iwi | Mah-tow-rah-nga-ah-e-we | Iwi-specific (tribal) knowledge |
| Mauri | Mouw-ree | Life force, essence |
| Mōhiotanga | Moh-he-o-ar-tah-nga | Acknowledgement, respect, awareness of potential |
| Pākehā | Pah-keh-hah | Non-Māori (European descent) New Zealander |
| Papatuanuku | Pah-pah-two-ah-nu-ku | Earth mother (Primal parent) |
| Pūrākau | Puh-rah-kouh | Oral record or history, often in story form |
| Ranginui | Rahng-e (as in letter e)-nu-e (as in letter e) | Sky father (Primal parent) |
| Rūnanga | Ru-nah-nga | Tribal council or governing board |
| Tane | Tah-neh | God of the forests; created the first human |
| Taniwha | Tah-knee-fah | Supernatural creatures in Māori legends, often taking the form of a serpent or water monster |
| Tangata whenua | Tah-nga-tah fe-nu-ah | People of the land |
| Taonga | Tah-or-nga | Treasure (noun), to be treasured (verb) |
| Te Ao Māori | Teh Owe Mah-or-ree | Māori worldview |
| Te Ao Marama | Teh Owe Mah-rah-mah | The world of light, the world we inhabit |
| Te Kore | Teh Kor-reh | The nothingness, the potential for life |
| Te Po | Teh Pore | The darkness, the night |
| Te taiao | Teh Tie-Owe | The natural world; the environment, including people |
| Tikanga | Tea-kah-nga | Customary practices, values, protocols |
| Tino rangatiratanga | Tea-nor Rah-nga-tea-rah-tah-nga | Self-determination |
| Ūkaipō | U (as in **cue**)-kay (as in kayak)-pore | Roots |
| Wairuatanga | Why-rue-ah-tah-nga | Spiritual dimension |
| Wānanga | Wah-nah-nga | Discussion |
| Whakakotahitanga | Far-kah-koh-tah-he-tah-nga | Respect for differences, ability to reach consensus, participatory inclusion in decision-making |
| Whakapapa | Far-kah-pah-pah | Ancestral genealogy, applicable to all elements of nature |
| Whakataukī | Far-kah-tow-key | Story or proverb |
| Whānau | Far-know | Family or close kin network |
| Whānaungatanga | Far-know-nga-tah-nga | Family connections |

a iwi (*iwi*-specific *mātauranga*), tribal ancestry, credibility and *iwi*-specific guardianship of tribal lands. Glossaries of Māori kupu, or words (Table 1), and key English terminologies used in this paper (Table 2) are provided for reference.

185

*Te Ao Māori* has, at its foundation, relationships between everything seen and unseen, humans and more-than humans, the natural and beyond-natural world, and in turn, shapes Māori ways of doing and living (Clapcott et al., 2018). Māori have been creating and revising their *mātauranga* since they first arrived to Aotearoa-NZ many centuries years ago (Hikuroa, 2017). After

settling, Māori formed distinct groups (about 40 *iwi* and hundreds of *hapū*), all of which built their identity from the surrounding mountains, lakes and rivers (Ruru, 2018). These tribal identities have implications for *mātauranga*

Table 2: Glossary of English terms (as used in this paper)

| Cultural association | The cultural uses and values associated with a landscape |
|---|---|
| Framework | Theoretical guides to research; methodolog |
| Geomorphic rights | Rights of a river with the status of legal personhood, understood from a geomorphic perspective |
| Indigenous knowledge | Knowledge generated by Indigenous peoples using Indigenous methods and usually including values, culture and worldview |
| Knowledge | Intellectual capital generated over time and carried through a range of channels including stories, songs, philosophies and teachings |
| Method | Acts by which research is conducted or specific research tool |
| Methodology | Philosophical approach to research or general research strategy |
| Model | Actionable guides to research; method |
| Science | The pursuit of knowledge according to the scientific method, and all of the knowledge generated using that method |
| Treaty of Waitangi | Aotearoa New Zealand's founding document; an agreement in Māori and English, made between Māori chiefs and the British Crown |
| Value | Guiding principles that support or enable acceptable actions |

**3.1.1** *Whakapapa* **and** *tikanga* **(—validity through ancestry)**

*Whakapapa* (ancestry) is the Māori way of understanding the world through genealogies (Forster, 2019). It links people to flora, fauna, mountains, rivers, oceans and lakes through an understanding that all of nature  descended from the *atua* (Māori gods) (Fig. 1; Harmsworth and Awatere, 2013). *Whakapapa* informs *tikanga* (cultural protocols and habits) which in turn informs how one should conduct their life (Graham, 2009).

[Figure]

**Figure 1. The pedigree of mankind in *Te Ao Māori. Modified from* Graham, 2009.**

*Whakapapa* is at the core of Indigenous Māori knowledge generation (Graham, 2009). *Whakapapa* legitimates Māori epistemologies within research and fosters credibility by establishing connections between researchers and research objectives, and by guiding research questions based on *tikanga* (Graham, 2009). By understanding that all things—both physical and metaphysical—are connected through genealogy (Hikuroa, 2017), it can be understood that *whakapapa* is a structured methodology for creating *mātauranga* (Royal, 1998). The relationships within *whakapapa* inform histories, stories, and interactions, which can be analysed in a Māori-centred way to create new knowledge (Fig. 2).

[Figure]

**Figure 2. Generation of Māori knowledge. *Modified from* Graham, 2009.**

**3.1.2 *Mātauranga* Māori (Māori knowledge)**

*Mātauranga* Māori is a detailed and dynamic way of knowing that has its *ūkaipō* (roots) in Māori ancestry (Paul-Burke et al., 2018). *Mātauranga* is a *taonga* (treasure) that is lived, practiced, tested, updated and that grows and develops as it is passed

from generation to generation. Based on Polynesian origins (Clapcott et al., 2018), Māori have been developing their *mātauranga* since their arrival to Aotearoa-NZ some 800-1000 years ago (Broughton et al., 2015). *Mātauranga* is not only knowledge, but is also a method of expressing knowledge through language, cultural practices, values, principles and ethics (Hikuroa, 2017; Paul-Burke et al., 2018). *Mātauranga taiao* (Māori environmental knowledge) is both traditional and contemporary and reflects the totality of Māori interactions with the environment during their occupation of Aotearoa-NZ (King et al., 2007).

*Mātauranga-a-iwi* provides local, place-based knowledge for an *iwi's* tribal area. This knowledge can provide intimate understandings of landscape dynamics and change through time. *Mātauranga-a-iwi* is informed directly by *whakapapa* (ancestry) because local landscape features are seen as kin through genealogical ties (Wilkinson and Macfarlane, in press; Ruru, 2018). The aim is to live with the environment in an intergenerationally sustainable way in which the landscape and its resources are respected as elders. Interacting with specific landscape features has generated and developed *mātauranga-a-iwi* and continues to refine local Indigenous knowledge.

**3.1.3 *Kaitiakitanga* (well-being of people and environment)**

In *Te Ao Māori*, *mana whenua* (traditional authorities) are the *kaitiaki* (guardians) of their lands, waters, and physical and cultural environments. *Kaitiakitanga* (guardianship) 
[revised manuscript text omitted]

**4.1.2 IBRLA (initiation, benefits, representation, legitimation, accountability)**

The IBRLA (initiation, benefits, representation, legitimation, accountability) framework is also an open-ended research framework that aims to ensure that Māori thinking and voice are included in research involving Māori (Bishop, 1996;

Table 3. The IBRLA framework. *Adapted from* Macfarlane and Macfarlane, 2018.

Table 3: The IBRLA framework. *Adapted from* Macfarlane and Macfarlane, 2018.

|   | Principle | Accountability Questions |
|---|-----------|--------------------------|
| I | Initiation | • Who conceptualised and initiated this research project?
 • How did Māori participate in the conceptualisation and initiation process?
 • How was the agreement to proceed with the research achieved? |
| B | Benefits | • How will the research (process and outcomes) accrue benefits for Māori?
 • How has information been shared with Māori about the intended benefits?
 • How will these benefits be determined and measured—and by whom? |
| R | Representation | • Whose ideas will be represented in the methodology, design and approach?
 • How will Māori thinking and knowledge be represented at all research phases?
 • How will this be monitored so that ongoing agreement/partnership is maintained? |
| L | Legitimation | • Who will legitimate the analysis and interpretation of information/research data?
 • How will Māori understandings be legitimately represented?
 • How will this be structured so that research fidelity is achieved/protected? |
| A | Accountability | • Who is accountable to whom—and in what ways?
 • How will on-going and mutual accountability be built into the research process?
 • How will this be monitored and evaluated to ensure safety for all stakeholders? |

Macfarlane and Macfarlane, 2018). It features a series of accountability questions within each component of the framework (Table 3). These questions are meant to guide researchers and help ensure that Māori knowledge is being included throughout the research project. These questions, such as "How did Māori participate in the conceptualisation and initiation process?" or "How will Māori thinking and knowledge be represented at all research phases?" hold researchers responsible for ensuring that Māori involvement and contribution is not only included but prioritised in the research. Principles of the Treaty of Waitangi—partnership, participation and protection—feature throughout the IBRLA framework. In this framework,   the accountability questions help ensure that *mātauranga* Māori is respected and upheld throughout the research process.

Just as the scientific method often encourages revisiting hypotheses, the IBRLA framework encourages researcher reflection during the concept design stage (similar to hypothesis formation and method development) through to the end of the research. The intent of IBRLA is to produce collaborative research stories (Bishop, 1996). This framework can provide a sense of researcher security when including Māori knowledge, while maintaining the integrity of the scientific method.

**4.1.3** *He Awa Whiria* **(a braided rivers approach)**

The *He Awa Whiria* framework is based on the imagery of  braided river systems (iconic landscape features throughout Aotearoa-NZ) and traditional woven baskets (Fig. 4).  A research project designed under the *He Awa Whiria*  framework has two streams, one representing  Western science and the other representing Māori knowledge. Like a braided river, the streams may diverge, converge, and meander, but ultimately, they both flow in the same direction and towards the same goal. The streams are accompanied by the metaphor of knowledge *kete* (baskets), which is inspired by the Māori *whakataukī* (saying/proverb) "*nā tō rourou, nā taku rourou,*— *ka ora ai te iwi*" ("with your food basket and my food basket, the people will thrive"). These symbols represent the weaving of Western science and Indigenous knowledge through a Māori worldview, in which the integrity and sovereignty of each is respected.

Throughout the duration of  a research project, the streams may wane or grow in strength, change directions, or even die out in places, as do the channels in a braided river. Both streams have the same objective, which is to provide balanced contributions to research outcomes. It is accepted that the streams may spend more time apart than together (Macfarlane and Macfarlane, 2018).  It is the researcher's role to manage how and when the two streams must converge, and when it is appropriate for them to diverge. It

480  is the also researcher's responsibility to make the moments of convergence times of learning. Ultimately, when research conclusions are drawn, the claimsy must be supported by both streamsrepresent co-creation of knowledge using both streams.

The *He Awa Whiria* methodology allows for flexibility within a research project. It recognises the benefits of both the Western science paradigm and *kaupapa* Māori principles and allows the research team to determine their own checks and
485  balances. It enables Western science to stay true to the scientific method. It also provides grounds for *mātauranga* Māori input to guide and focus the Western science analysis. Wilkinson and Macfarlane (in press) demonstrate that the *He Awa Whiria* method framework can be applied to geomorphic studies by allowing the two knowledge two streams to operate both independently and collaboratively. The *He Awa Whiria* methodology supports a culturally responsible and responsive approach to research and allows for variable approaches to research depending on the specific topic (Macfarlane et al., 2015; Macfarlane
490  and Macfarlane, 2018). Methodological adaptability is essential for conducting research with Māori, because different Māori groups will have different values, priorities and interests when it comes to pursuing research questions.

[Figure]

**Figure 4. The *He Awa Whiria* framework. The blue lines represent knowledge exchange and development as the two streams converge and re-converge throughout the research programme. *
[revised manuscript text omitted]

inclusion of *iwi*-identified cultural, environmental, and research values in geomorphic investigations. Model selection, like

650 framework selection, will depend on the research questions at hand and must be done on a case-by-case basis as a joint decision

between the Māori community from whom the *mātauranga* is sourced. Regardless of how Māori knowledge is included, -it is

ideal to have a Māori researcher on the project leadership team  to minimise risk of cultural

misrepresentation or appropriation of knowledge.  *iwi*

 *tangata whenua*

655

**5.3 Guiding resources for initiating projects in Aotearoa-NZ**

As previously discussed, many *iwi* (tribes) and *hapū* (sub-tribes) in Aotearoa-NZ have published *iwi* management plans or *iwi*

environmental management plans that can outline research priorities for scientists (Saunders, 2017). Many IMPs contain

information specifically relevant to geomorphologists. For example, most IMPS discuss *iwi* goals for minimum river flows

660 and mitigating flood hazards (e.g., Tipa et al., 2014). Other *iwi* environmental management plans have sections with specific goals for rivers (e.g., Waikato-Tainui Te Kauhanganui Incorporated, 2013) or catchments (e.g., Te Rūnanga o Kaikōura et al., 2005; Jolly and Ngā Papatipu Rūnanga Working Group, 2013). Most IMPs focus on improving the *mauri* (life force, vitality) of tribal landscapes over which the authoring organisation (typically a *rūnanga*, tribal council) possesses *kaitiakitanga* (guardianship). Researchers can use information outlined in IMPs and IEMPs to identify the appropriate research leadership

665 team and select the appropriate research framework.

Research funding guidelines for projects that aim to include *mātauranga* Māori alongside Western science can be found through Aotearoa-NZ's Ministry of Research, Science and Technology. Specifically, the Ministry of Business, Innovation and Education (MBIE) operates Te Punaha Hihiko, the Vision Mātauranga Capability Fund, which provides guidelines for research projects in its application process (Ministry of Research, Science and Technology, 2007). The Marsden

670 Fund, through the Royal Society of New Zealand, also provides for how proposals should include consideration of Māori involvement in research (Royal Society Te Apārangi, n.d.).

Many universities and research organisations have *iwi* engagement and support teams. These teams are an excellent resources for gaining guidance on identifying the best *tangata whenua* team for research needs, and how to appropriately engage with that *iwi* or *hapū*. In Aotearoa-NZ, the universities and CRIs, in particular, have excellent resources for connecting

675 researchers with Māori. We recommend early and, ideally, regular interaction with these resource groups. It is worth noting that an argument exists to make Māori representation on research project teams mandatory, but ultimately, forcing Māori involvement runs the risk of perpetuating colonizing practices. Instead, we maintain that the Māori community should decide how much—or how little—they wish to contribute to research projects. Engagement and support teams will be able to advise on this subject.

680 **6. Lessons for the international geomorphology community**

Indigenous knowledge around the globe is a valid source of information because it has endured for generations, keeping populations alive and securing their livelihoods. Moreover, Indigenous knowledge has been shown to be accurate and precise (Hikuroa, 2017). In this section, we outline some direct benefits of including Indigenous knowledge in geomorphic research, discuss how the frameworks detailed in this review can be adapted for use outside of Aotearoa-NZ, and discuss how Indigenous

685 knowledge and geomorphic research can and are working together to inform sustainability policy and legislation.

**6.1 Direct benefits to geomorphology**

A clear benefit to geomorphology is the temporal extension of observations of geomorphic events into pre-history. The 400-year volcanic record discussed by Swanson (2008) and the cycles of flood and channel avulsion evaluated by Hikuroa (2017) indicate that Indigenous knowledge can bolster scientifically-investigated geomorphic histories. King and Goff (2006)

690 further demonstrated that Māori oral histories frequently discuss multiple geomorphic phenomena happening in tandem or as

cascading events. Recognition of the interconnectedness of landscape processes is a common theme in many Indigenous societies (Riggs, 2005) and this recognition has resulted in a way of life that responds to, interacts with and learns from concurrent or cascading suites of local landscape processes.

Another key benefit of including Indigenous knowledge in geomorphic endeavours is the opportunity to co-create new approaches to research that build holistic and more complete understandings of landscape processes. Contemporary Indigenous knowledge and narratives can provide signposts for initiating and conducting geomorphic research by indicating geographic areas or research questions that are of interest to Indigenous groups. The concept of 'ethnogeomorphology' (Wilcock and Brierley, 2012; Wilcock et al., 2013) draws upon modern Indigenous knowledge and relationships with landscapes to guide geomorphic research questions and methodologies. The dynamic and adaptive nature of Indigenous knowledge generation (Berkes, 2009) has the potential to influence adaptive research methods, which in turn, have the potential to generate robust data collection with information from a variety of sources.

A prime example of how adaptive research methods incorporating Indigenous knowledge can provide significant contributions to geomorphic research is the New Zealand Palaeotsunami Database. The database aims to catalogue all tsunamis that occurred prior to written historical records and uses three types of evidence to identify palaeotsunami events: sedimentary/artefactual ("primary"), geomorphic ("secondary") and anthropological/*pūrākau* ("cultural") (Goff, 2008; New Zealand Palaeotsunami Database, 2017). The cultural information allowed the database compilers to better constrain the age of palaeotsunami events by dating archaeological sites that were associated with the cultural information (Goff, 2008). A typical prehistoric Māori response to big waves was to abandon coastal settlements and move to higher elevations (Goff and Chagué-Goff, 2015). Cultural knowledge of the locations of abandoned sites allowed researchers to conduct archaeological investigations and date the time at which such sites had been occupied, thus providing a well-constrained date for the tsunami event. Māori *pūrākau* (oral histories/stories) often provide even more detailed information (McFadgen and Goff, 2007). Stories of *taniwha* (supernatural creatures in Māori mythology) may indicate big wave events that wreaked havoc on coastal communities, causing changes in settlement and local geomorphology (King and Goff, 2010; Goff and Chagué-Goff, 2015). Currently, cultural information is included for 14% of recorded tsunami events in the database, most of which have come from *pūrākau* ("New Zealand Palaeotsunami Database," 2017). The cultural information, alongside geomorphic and sedimentary information, provide key data for the generation of a robust and comprehensive palaeotsunami database for Aotearoa New Zealand (Goff et al., 2010).

**6.2 International application of Aotearoa-NZ bicultural research frameworks and models**

Indigenous communities around the world share many fundamental principles, including their interconnectedness with and inseparability from nature (Salmón, 2000; Wambrauw and Morgan, 2016). Other cultural values, such as environmental stewardship and sustainability, are also common Indigenous values that guide ways of living and ways of knowing. Common

values among Indigenous cultures enable and encourage transferability of established frameworks outside of the place where
725   they have been developed. The three theoretical frameworks discussed in this review—*He Poutama Whakamana*, IBRLA and
*He Awa Whiria*—can potentially be applied outside of Aotearoa-NZ, due to their flexible nature and adaptability for different
research groups and purposes. Likewise, the value-based models—the *Mauri* MModel, the CFPS and SAM—can be modified
to incorporate Indigenous values and priorities outside of the Aotearoa-NZ context, because the models are created specified
with Indigenous groups on a case-by-case basis. Indigenous groups anywhere can specifyidentify which values they consider
730   essential for the frameworks and models.

The *Mauri* MModel, developed in Aotearoa-NZ, has been successfully applied in Papua, Indonesia to evaluate the
potential effects of a new agricultural development scheme in the Merauke regency, in the lowlands of Papua (Wambrauw and
Morgan, 2014, 2016). Due to its ability to incorporate Indigenous and Western values, the *Mauri* MModel was deemed an
appropriate tool to assess the potential environmental and cultural impacts of the development scheme. The first step to
735   successfully applying the model was to understand the new context in which it would be used. After confirming the *Mauri*
MModel would be appropriate, stakeholders for the project were selected, which included the Malind Anim Indigenous
peoples. The *Mauri* MModel was adjusted to have a minimum value of -3 and a maximum value of +3 (rather than -2 and +2,
respectively), based on local values and requirements. The results from using the *Mauri* MModel indicated that the cultural
values associated with the site would be denigrated if the development scheme proceeded. The *Mauri* MModel provided semi-
740   quantitative evidence that the development scheme would have serious negative impacts on the Malind Anim.

It is challenging to review the applicability of Aotearoa-NZ frameworks and models to international geomorphic
research because, to our knowledge, there are extremely few studies that explicitly use the frameworks tools to conduct
geomorphic research outside of Aotearoa-NZ. However, we believe that there is great potential for these frameworks and
models to be adapted outside of Aotearoa-NZ, or for these tools to act as inspiration for the generation of new frameworks and
745   models for use with Indigenous groups in other parts of the world. The case of using the *Mauri* MModel 
[revised manuscript text omitted]

2013.

935    Kauffman, C.M. and Martin, P.L.: Constructing Rights of Nature Norms in the US, Ecuador, and New Zealand, Global

Environmental Politics, 18, 43-62, doi:10.1162/glep_a_00481, 2018.

Kelman, I., Mercer, J., and Gaillard, J.C.: Indigenous knowledge and disaster risk reduction, Geography, 97,

https://search.proquest.com/docview/1459729135?pq-origsite=gscholar (accessed October 2019), 2012.

Kharusi, N.S., and Salman, A.: In Search of Water: Hydrological Terms in Oman's Toponyms, Names, 63, 16–29,

940    doi:10.1179/0027773814Z.00000000094, 2015.

King, D. and Goff, J.: Maori environmental knowledge in natural hazards management and mitigation, National Institute of

Water and Atmospheric Research Ltd GNS05301-1, 85 pp., 2006.

King, D.N. and Goff, J.R.: Benefitting from differences in knowledge, practice and belief: Māori oral traditions and natural

hazards science, Natural Hazards and Earth System Sciences, 10, 1927–1940, doi:https://doi.org/10.5194/nhess-10-

945    1927-2010, 2010.

King, D., Goff, J., and Skipper, A.: Māori environmental knowledge and natural hazards in Aotearoa-New Zealand, Journal

of the Royal Society of New Zealand, 37, https://www.tandfonline.com/doi/abs/10.1080/03014220709510536

(accessed June 2019), 2007.

King, D., Shaw, W., Meihana, P., and Goff, J.: Maori oral histories and the recurring impact of tsunamis in Aotearoa-New

950    Zealand, Natural Hazards and Earth System Sciences, 18, 907-919, https://doi.org/10.5194/nhess-18-907-2018,

2018.

Londono, S.C., Garzon, C., Brandt, E., Semken, S., and Makuritofe, V.: Ethnogeology in Amazonia: Surface-water systems

in the Colombian Amazon, from perspectives of Uitoto traditional knowledge and mainstream hydrology, in

Geological Society of America Special Papers, Geological Society of America, 520, 221–232,

955    doi:10.1130/2016.2520(20), 2016.

Macfarlane, S. and Macfarlane, A.H.: Toitū Te Mātauranga: Valuing Culturally Inclusive Research in Contemporary Times :

a Position Paper Prepared Under the Auspices of the Māori Research Laboratory, Te Rū Rangahau, in Conjunction

with the Child Wellbeing Institute at the University of Canterbury: University of Canterbury, 8 pp., 2018.

Macfarlane, S., Macfarlane, A., and Gillon, G.: Sharing the food baskets of knowledge: Creating space for a blending of

960    streams, in: Sociocultural realities: Exploring new horizons, edited by: Macfarlane, A., Macfarlane, S., and Webber,

M., Christchurch, NZ, Canterbury University Press, 2015.

Marsden, M., Goodall, A., Palmer, D.: Ministry for the Environment and Core Group on Resource Management Law Reform (N.Z.) Resource management law reform : part A, The natural world and natural resources, Maori value systems & perspectives : part B, Water resources and the Kai Tahu claim. Ministry for the Environment, Wellington, 1989.

965

Massey, C.I., Townsend, D., Dellow, S., Lukovic, B., Rosser, B., Archibald, G., Villeneuve, M., Davidson, J., Jones, K., Morgenstern, R., Strong, D., Lyndsell, B., Tunnicliffe, J., Carey, J., and McColl, S..: Kaikoura Earthquake Short Term Project: Landslide inventory and landslide dam assessments, GNS Science report; 2018/19, 45 pp., 2018.

Maxwell, K.H., Ratana, K., Davies, K.K., Taiapa, C., and Awatere, S.: Navigating towards marine co-management with

970
Indigenous communities on-board the Waka-Taurua, Marine Policy, 111, 4 pp., doi:10.1016/j.marpol.2019.103722, 2020.

[revised manuscript text omitted]